# Direct binding of Cdt2 to PCNA is important for targeting the CRL4$^{Cdt2}$ E3 ligase activity to Cdt1

Akiyo Hayashi[1] , Nickolaos Nikiforos Giakoumakis[2],*, Tatjana Heidebrecht[3],* , Takashi Ishii[1], Andreas Panagopoulos[2], Christophe Caillat[3], Michiyo Takahara[1], Richard G Hibbert[3], Naohiro Suenaga[1], Magda Stadnik-Spiewak[3] , Tatsuro Takahashi[5] , Yasushi Shiomi[1], Stavros Taraviras[4], Eleonore von Castelmur[3] , Zoi Lygerou[2] , Anastassis Perrakis[3] , Hideo Nishitani[1]

**The CRL4$^{Cdt2}$ ubiquitin ligase complex is an essential regulator of cell-cycle progression and genome stability, ubiquitinating substrates such as p21, Set8, and Cdt1, via a display of substrate degrons on proliferating cell nuclear antigens (PCNAs). Here, we examine the hierarchy of the ligase and substrate recruitment kinetics onto PCNA at sites of DNA replication. We demonstrate that the C-terminal end of Cdt2 bears a PCNA interaction protein motif (PIP box, Cdt2$^{PIP}$), which is necessary and sufficient for the binding of Cdt2 to PCNA. Cdt2$^{PIP}$ binds PCNA directly with high affinity, two orders of magnitude tighter than the PIP box of Cdt1. X-ray crystallographic structures of PCNA bound to Cdt2$^{PIP}$ and Cdt1$^{PIP}$ show that the peptides occupy all three binding sites of the trimeric PCNA ring. Mutating Cdt2$^{PIP}$ weakens the interaction with PCNA, rendering CRL4$^{Cdt2}$ less effective in Cdt1 ubiquitination and leading to defects in Cdt1 degradation. The molecular mechanism we present suggests a new paradigm for bringing substrates to the CRL4-type ligase, where the substrate receptor and substrates bind to a common multivalent docking platform to enable subsequent ubiquitination.**

## Introduction

The integrity of genomic information is maintained in the cell cycle by faithful replication during the S phase and segregation of duplicated chromosomes during mitosis, which is critical for proper cell reproduction, cell function, and cell survival. In addition, cells are continuously challenged by genotoxic agents and environmental stress, and have complex mechanisms to activate DNA damage checkpoints, prevent cell-cycle progression, and repair the damaged DNA (Hoeijmakers, 2001; Branzei & Foiani, 2010). Many of the cell cycle transition events, as well as responses to DNA damage, are driven by E3 Cullin-RING ubiquitin Ligases (CRLs) that catalyse the ubiquitination and destruction of specific protein targets. Such cell cycle–regulated E3 ligases include CRL1$^{Fbox}$ and CRL4$^{DCAF}$, which target many substrates crucial for cell cycle regulation and DNA damage responses (Cardozo & Pagano, 2004; Petroski & Deshaies, 2005; Jackson & Xiong, 2009). These CRLs comprise a scaffolding protein (cullin 1 or cullin 4 [Cul4]), an adapter protein (Skp1 and DDB1, respectively), and a RING domain protein that interacts with the E2 (such as Rbx1 or Rbx2). Finally, CRL1 and CRL4 ligases contain either an F-box or DCAF substrate recognition factor (SRF, or substrate receptor), respectively, responsible for interacting with the substrate and targeting it for ubiquitination. F-box proteins in CRL1, such as Fbw7 or β-TRCP, recognize specific degrons in substrates that often contain phosphorylated residues, whereas CRL4 include DCAFs such as DDB2, which directly recognizes UV-damaged DNA (Scrima et al, 2008).

The CRL4$^{Cdt2}$ ligase uses Cdt2 as the SRF, and functions both during the S phase and after DNA damage (Abbas & Dutta, 2011; Havens & Walter, 2011; Sakaguchi et al, 2012; Stathopoulou et al, 2012). Cdt2, targets substrates such as p21 and Set8, and the DNA replication licensing factor Cdt1 for ubiquitin-mediated proteolysis, both in S phase and following DNA damage (Abbas et al, 2008; Kim et al, 2008; Nishitani et al, 2008; Centore et al, 2010; Oda et al, 2010; Tardat et al, 2010; Jorgensen et al, 2011). In addition, an increasing number of Cdt2 target proteins have been identified, including thymine DNA glycosylase, Cdc6, the DNA polymerase δ subunit p12 (Terai et al, 2013; Clijsters & Wolthuis, 2014; Shibata et al, 2014; Slenn et al, 2014), and xeroderma pigmentosum group G (XPG), a

[1]Graduate School of Life Science, University of Hyogo, Kamigori, Japan   [2]Department of Biology, School of Medicine, University of Patras, Patras, Greece   [3]Department of Biochemistry, Netherlands Cancer Institute, Amsterdam, The Netherlands   [4]Department of Physiology, School of Medicine, University of Patras, Patras, Greece   [5]Faculty of Science, Kyushu University, Fukuoka, Japan

Correspondence: lygerou@med.upatras.gr; a.perrakis@nki.nl; hideon@sci.u-hyogo.ac.jp
Takashi Ishii's present address is Department of Biochemistry, Fukuoka Dental College, Fukuoka, Japan
Christophe Caillat's present address is CNRS (The National Centre for Scientific Research) , UVHCI, 71 avenue des Martyrs, Grenoble, France
Richard G Hibbert's present address is Genmab B.V., Utrecht, The Netherlands
*These authors contributed equally to this work

structure-specific repair endonuclease of the nucleotide excision repair pathway (Han et al, 2015).

Cdt1 and Cdt2 were originally identified as Cdc10-dependent transcript 1 and 2 in fission yeast, but have no sequence similarity (Hofmann & Beach, 1994). Cdt1 has a critical role in establishing the DNA replication licensing complex in the G1 phase: it associates with chromatin through the origin recognition complex and operates together with Cdc6 to load the MCM2-7 complex onto chromatin, thereby licensing DNA for replication (Bell & Dutta, 2002; Diffley, 2004; Nishitani & Lygerou, 2004; Blow & Dutta, 2005; Tsakraklides & Bell, 2010; Symeonidou et al, 2012).

Preventing re-licensing of replicated regions is essential (Blow & Dutta, 2005; Arias & Walter, 2007). One of the mechanisms to achieve this is by CRL1$^{Skp2}$ and CRL4$^{Cdt2}$ redundantly mediating Cdt1 destruction in higher organisms. CRL1$^{Skp2}$ (also known as SCF$^{Skp2}$) recognizes a phospho-degron motif on Cdt1 that is created at the initiation of S phase by CDKs (Li et al, 2003; Sugimoto et al, 2004; Nishitani et al, 2006). In contrast, CRL4$^{Cdt2}$ recognizes Cdt1 when bound to the proliferating cell nuclear antigen (PCNA) trimer, through a binding motif (PIP box) in its N-terminal end (Arias & Walter, 2006; He et al, 2006; Higa et al, 2006; Jin et al, 2006; Nishitani et al, 2006; Ralph et al, 2006; Sansam et al, 2006; Senga et al, 2006; Kim & Kipreos, 2007). Both initiation of DNA replication and DNA damage trigger PCNA loading onto chromatin and Cdt1 association with PCNA through its PIP box (Arias & Walter, 2006; Havens & Walter, 2009; Raman et al, 2011; Shiomi et al, 2012). DNA damage–induced degradation of Cdt1 and other substrates appears to facilitate repair (Mansilla et al, 2013; Tsanov et al, 2014; Tanaka et al, 2017).

Cdt2 recruitment onto chromatin is not fully characterized: recruitment through the Cdt1 PIP box bound to PCNA and a specific basic residue downstream of the Cdt1 PIP box (Havens & Walter, 2009; Michishita et al, 2011; Havens et al, 2012) or independently of Cdt1 (Roukos et al, 2011) has been reported. Following CRL4$^{Cdt2}$-mediated ubiquitination, Cdt1 is degraded, thus blocking further licensing. The N-terminal region of Cdt2 contains a WD40 repeat domain, predicted to form a substrate-recognizing propeller structure similar to the one shown for DDB2 (Scrima et al, 2008; Havens & Walter, 2011). In analogy with DDB2 (Fischer et al, 2011), the N-terminal domain of Cdt2 should bind to the DDB1 WD40 repeat domains, β-propeller A and β-propeller C, on one side. The other side would be expected to recognize the substrate, in analogy to DDB2. Higher eukaryotic Cdt2 proteins have an extended C-terminal region, not present in fission yeast. In Xenopus, the C-terminal domain of Cdt2 binds to PCNA and is important for the turnover of the Xic1 cyclin kinase inhibitor (Kim et al, 2010). Recently, we reported that Cdt2 mutated at multiple CDK consensus phosphorylation sites colocalized with PCNA throughout the S phase even when most of the substrates were degraded, also suggesting that Cdt2 interacts with PCNA independent of its substrates (Nukina et al, 2018).

To understand how Cdt2 recognizes PCNA and localizes CRL4$^{Cdt2}$ activity during the S phase and following UV irradiation, we investigated the role of Cdt2 domains in localization and ubiquitination. Unexpectedly, we found a PIP box in the C-terminal end of Cdt2, and showed that it directly mediates interaction of Cdt2 with PCNA. Both Cdt1 and Cdt2 PIP box peptides bind the PCNA binding pocket in a similar manner, but Cdt2 has a significantly higher affinity for PCNA. We suggest that Cdt2 and Cdt1 could simultaneously recognize different subunits of the PCNA trimer, and we put forward a new paradigm for localizing E3 ligase activity onto the PCNA docking platform, allowing simultaneous docking of substrates in the same platform, enabling ubiquitination by proximity.

# Results

## Cdt2 is stably bound to UV-damaged sites independently of Cdt1

CRL4$^{Cdt2}$ recognizes Cdt1 bound on PCNA and recruitment of CRL4$^{Cdt2}$ to damaged chromatin was reported to require Cdt1 in Xenopus extracts (Havens & Walter, 2009). In contrast, experiments in live human cells suggested that Cdt2 was recruited to DNA damage sites independently of Cdt1 (Roukos et al, 2011). To verify whether Cdt2 binding to damaged sites requires Cdt1, we performed a time-course analysis of Cdt2 recruitment to sites of localized UV-C irradiation in control and Cdt1 depleted HeLa cells. Cells were exposed to UV-C through micropore filters, and locally induced damage was detected by the staining of cyclobutane pyrimidine dimers (CPDs). In control cells, Cdt1 was rapidly recruited to damaged sites (Fig 1A, 10 min) and proteolysed by 30 min (Fig 1A and B), consistent with earlier studies (Ishii et al, 2010; Roukos et al, 2011). Cdt2 was recruited at sites of damage by 10 min whereas cells showing Cdt2 recruitment increased at subsequent time points (30 and 45 min post-irradiation) even-though Cdt1 was degraded. Depletion of Cdt1 by siRNA had no effect on either the kinetics or the extent of Cdt2 recruitment to sites of UV-C damage (Fig 1A and B). These results show that Cdt2 is recruited to UV-C–irradiated sites independently of Cdt1 and remains there long after Cdt1 proteolysis.

To assess the binding properties of Cdt2 at sites of UV-C damage in the presence and absence of Cdt1, fluorescence recovery after photobleaching (FRAP) was used. MCF7 cells transiently expressing Cdt1, Cdt2, or PCNA tagged with GFP were first analysed in the presence and absence of local UV-C irradiation (Figs 1C and S1A–C). All factors exhibit fast recovery in the absence of damage (Fig S1A and C), consistent with transient interactions (Xouri et al, 2007b). Following recruitment to the sites of damage, PCNA shows a significant immobile fraction, as expected for stable binding, whereas Cdt1 shows dynamic interactions at UV-C–damaged sites (Roukos et al, 2011; Rapsomaniki et al, 2015). Cdt2 has a slower fluorescence recovery rate at the site of damage than Cdt1, with a half-recovery time of 0.83 s and an immobile fraction of 16%, underlining long-term association at UV-C–damaged sites. To assess whether Cdt2 binding to sites of damage is affected by the presence of Cdt1, Cdt2 kinetics were assessed by FRAP in MCF7 cells synchronized in G1, depleted of Cdt1 and locally UV-C irradiated. As shown in Fig 1D and in Fig S1D and E, the binding kinetics of Cdt2 at the sites of damage remain the same despite the absence of Cdt1.

We conclude that Cdt2 binds to sites of damage stably independently of Cdt1.

## The C-terminal part of Cdt2 is required for recruitment to DNA damage sites

The above results suggested that Cdt2 contains a domain that mediates its association with UV-damaged sites independently

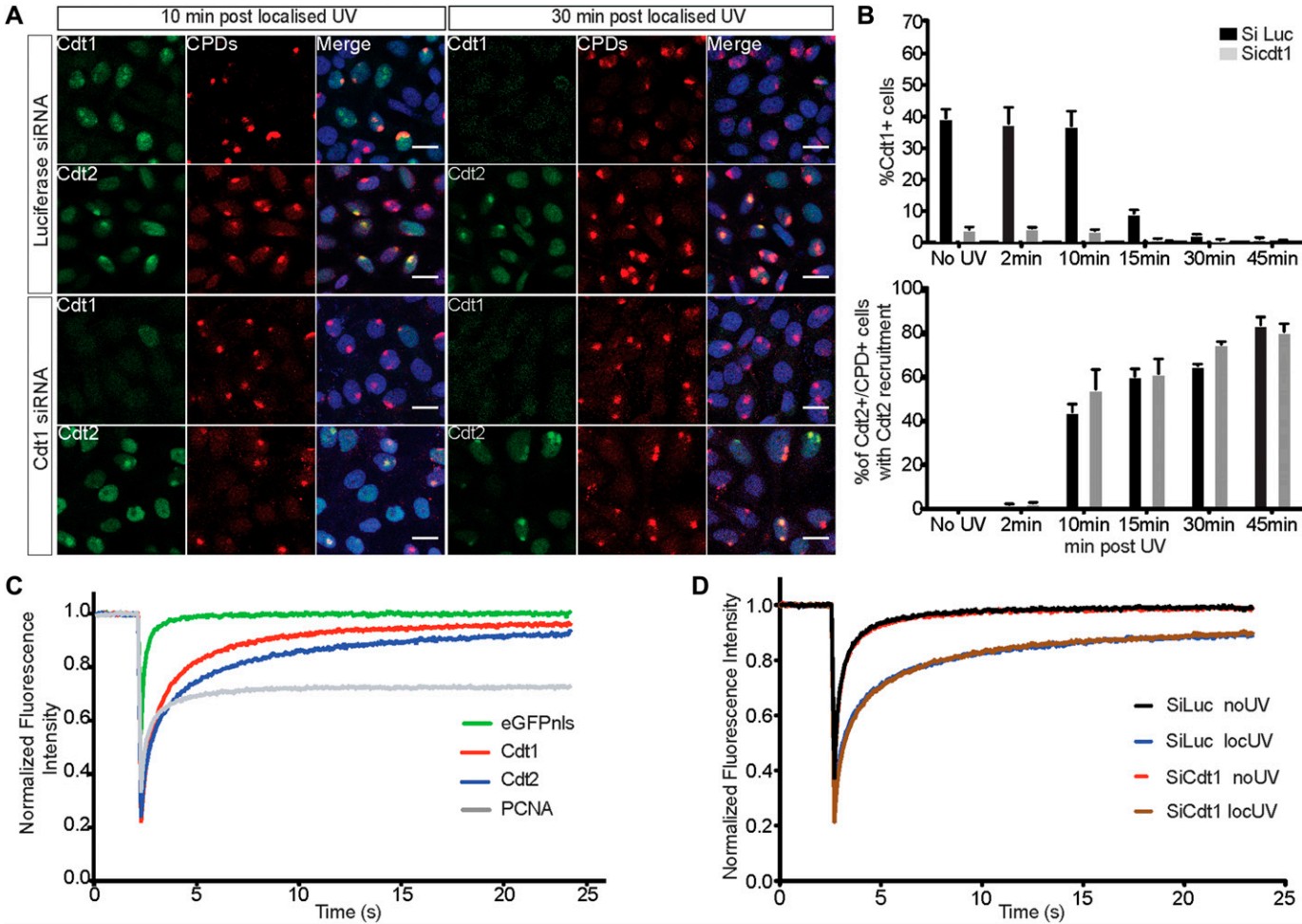

**Figure 1. Cdt2 binds stably to UV-damaged sites independently of Cdt1.**
**(A)** HeLa cells were transfected with siRNAs targeted to Cdt1 (siCdt1) or control (siLuc), and locally UV-C irradiated (50 J/m$^2$ through a micropore filter with 5-$\mu$m diameter pores), fixed at the indicated time points, and stained with antibodies against CPDs and Cdt1 or Cdt2. Nuclei were stained with Draq5. Scale bars: 20$\mu$m. **(B)** Percentage of cells positive for Cdt1 (top) and percentage of cells positive for CPDs and Cdt2 which show Cdt2 recruitment to sites of damage (bottom) are plotted (mean of two independent experiments with SD). **(C)** MCF7 cells were transfected with the indicated GFP-tagged plasmids and 24 h later they were locally UV-C irradiated (50 J/m$^2$). 20 min following irradiation, FRAP experiments were conducted at the site of damage and in untreated cells (Fig S1). FRAP data were analysed with easyFRAP. Mean normalized fluorescence intensities at the site of damage as a function of time following photobleaching are shown. **(D)** MCF7 cells were synchronized in the S phase by treatment with 2-mM thymidine for 24 h. Upon release, cells were transfected with siCdt1 or control siLuc in parallel with a plasmid expressing GFP-tagged Cdt2. Cells were then synchronised in the M phase by treatment with 50 ng/ml nocodazole for 12 h, released into G1 for 5 h, and locally UV-C irradiated (50 J/m$^2$). 20 min after irradiation, the cells were analysed by FRAP. Mean normalized fluorescence recovery curves at the sites of damage are shown.

of Cdt1. Human Cdt2 is a 730 amino acid polypeptide, with an N-terminal WD40 domain predicted to form a substrate receptor and a long C-terminal domain (Fig 2A). Constructs expressing Cdt2$^{1–417}$, which contains the N-terminal WD40 domain, the C-terminal Cdt2$^{390–730}$, and the full-length Cdt2$^{1–730}$ as a control fused with GFP were transiently expressed in MCF7 cells, followed by localized UV-C irradiation (Fig 2B and C). Cdt2$^{1–730}$ was robustly detected at DNA-damage sites, as previously reported. The C-terminal Cdt2$^{390–730}$ construct was also efficiently recruited to DNA-damage sites. On the contrary, there was no evidence for recruitment of Cdt2$^{1–417}$ to sites of damage. This suggests that the C-terminal region of Cdt2, but not the predicted N-terminal substrate receptor domain, is important for its recruitment to the sites of damage.

**The C-terminal part of Cdt2 is required for interaction with PCNA and has a PIP box in its C-terminal end**

Consistent with the observation that the C-terminal region of Cdt2 is important for recruitment to the CPD-stained sites where PCNA also localizes, we identified PCNA as a Cdt2 C-terminus interacting protein in a yeast two-hybrid screening (Fig S2). To confirm the interaction with PCNA, we transiently transfected 3FLAG-tagged Cdt2$^{1–730}$, Cdt2$^{1–417}$, and Cdt2$^{390–730}$ into HEK293T cells. Before cell lysate preparation, cells were cross-linked. Immunoprecipitation (IP) experiments using $\alpha$-FLAG antibody showed that the C-terminal half, but not the N-terminal half, interacts with PCNA (Fig 3A). The Cdt2$^{1–417}$ domain, although it lost most of its affinity for PCNA, is still able to bind DDB1 as expected. In contrast, the Cdt2$^{390–730}$ domain,

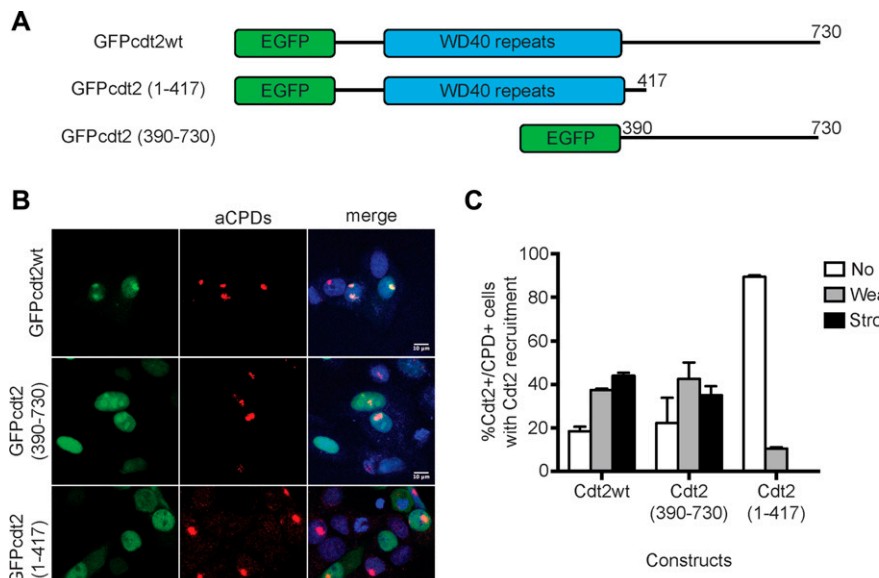

**Figure 2. The C-terminus of Cdt2 mediates Cdt2 recruitment to sites of damage.**
**(A)** A schematic drawing of the domain structure of Cdt2 and constructs used. **(B)** The C-terminal, but not the N-terminal, part of Cdt2 mediates recruitment to DNA damage sites. MCF7 cells transfected with the indicated plasmids were locally UV-C irradiated (50 J/m² through 5 µm pore diameter micropore filters), and recruitment of each Cdt2 construct to sites of damage was examined by staining with α-CPD antibody and GFP signal. **(C)** The percentage of CPD and Cdt2-positive cells which showed strong, weak, or no Cdt2 signal at CPD sites is plotted (mean of two independent experiments with SD).

although it still binds to PCNA, has lost its ability to bind DDB1, and therefore to form a functional CRL4 complex.

To narrow the region of Cdt2 required for interaction with PCNA, we made a series of additional constructs fused to a triple FLAG tag. We confirmed that when the C-terminal 30 amino acids were deleted in the Cdt2$^{1–700}$ construct, this was sufficient to lead to a loss of interaction with PCNA (Fig 3B). Consistently, the C-terminal 130 amino acids alone, Cdt2$^{600–730}$, were sufficient to mediate the interaction with PCNA.

We postulated that the C-terminal part of Cdt2 directly interacts with PCNA via a PCNA interaction protein (PIP) box; a common motif found in PCNA-dependent DNA replication and repair factors. We used PROSITE to search the UniProtKB protein sequence database for human sequences that contained consensus PIP boxes ([KMQ]-x-x-[ILMV]-x-x-[FY]-[FY]). Our search revealed a PIP box sequence from amino acids 706 to 713 in the C-terminus of Cdt2. This motif was well conserved in vertebrates and in *Drosophila* (Fig 3C). We then generated a triple-mutant where the three conserved hydrophobic residues I709, Y712, and F713, also known to interact with PCNA in structures of PIP box proteins bound to PCNA, were all mutated to alanine, yielding Cdt2$^{PIP-3A}$.

Introducing the Cdt2$^{PIP-3A}$ construct fused to a C-terminal FLAG tag to cells and subsequent IP experiments with anti-FLAG antibody after cross-linking, showed that the interaction of the Cdt2$^{PIP-3A}$ construct with PCNA was lost almost entirely (Fig 3D, UV - ), without significantly affecting interaction with Cul4A and DDB1. To confirm the interaction, we performed the reverse IP experiment. Myc-tagged PCNA was co-transfected with Cdt2$^{WT}$ or Cdt2$^{PIP-3A}$, and we performed an IP with anti-Myc antibody. Cdt2$^{WT}$ was co-precipitated with Myc-PCNA, but Cdt2$^{PIP-3A}$ was not detected in the precipitates (Fig 3E).

Next, we examined the interaction of CRL4$^{Cdt2}$ ligase with its target protein Cdt1, using HEK293 cells alone and cells transfected with Cdt2$^{WT}$ or Cdt2$^{PIP-3A}$, with or without UV irradiation. Using an anti-FLAG antibody resin to precipitate Cdt2$^{WT}$, both PCNA and Cdt1 were precipitated, and their amounts were notably increased after UV irradiation (Fig 3D). In contrast, neither Cdt1 nor PCNA co-precipitated well with Cdt2$^{PIP-3A}$, before or after UV irradiation. These results suggested that the PIP box of Cdt2 was required for forming a stable complex with its substrates on PCNA.

### A C-terminal PIP box in Cdt2 directly interacts with PCNA with high affinity

To confirm that this PIP box motif in Cdt2 interacts directly with PCNA, and to quantify that interaction, we synthesized the peptide corresponding to the human Cdt2 PIP box (704-717) (SSMRKICTYFHRKS) with a carboxytetramethylrhodamine fluorescent label at the N-terminus (Cdt2$^{PIP}$) and characterized its binding to recombinant PCNA, by fluorescence polarization (FP). Cdt2$^{PIP}$ binds PCNA with high affinity (57 ± 3 nM, Fig 4A). The corresponding Cdt2$^{PIP-3A}$ peptide showed practically no detectable binding to PCNA in the same assay (Fig 4A), suggesting that the Cdt2 PIP box binds PCNA in a manner very similar to other known PCNA complexes with PIP box peptides.

As Cdt1 is recruited to PCNA on chromatin during the S phase and after DNA damage through the Cdt1 N-terminal PIP box (residues 3–10) (Havens & Walter, 2009; Ishii et al, 2010; Roukos et al, 2011), we then synthesized a Cdt1$^{PIP}$-TAMRA (MEQRRVTDFFARRR) peptide, to compare its affinity with PCNA. Remarkably, the affinity of the Cdt1$^{PIP}$ peptide to PCNA (7,200 ± 200 nM, Fig 4A) is two orders of magnitude weaker than that for the Cdt2$^{PIP}$, and similar to what is reported for other peptides derived, for example, from the pol-δ p66 and FEN1 (1–60 µM) (Bruning & Shamoo, 2004). The affinity of the Cdt2$^{PIP}$ peptide for PCNA is similar to that reported for the tightly binding p21 PIP box peptide (50–85 nM) (Bruning & Shamoo, 2004; De Biasio et al, 2012), as it was confirmed in our assays (RRQTSMTDFYHSKR, 36 ± 2 nM, Fig 4A).

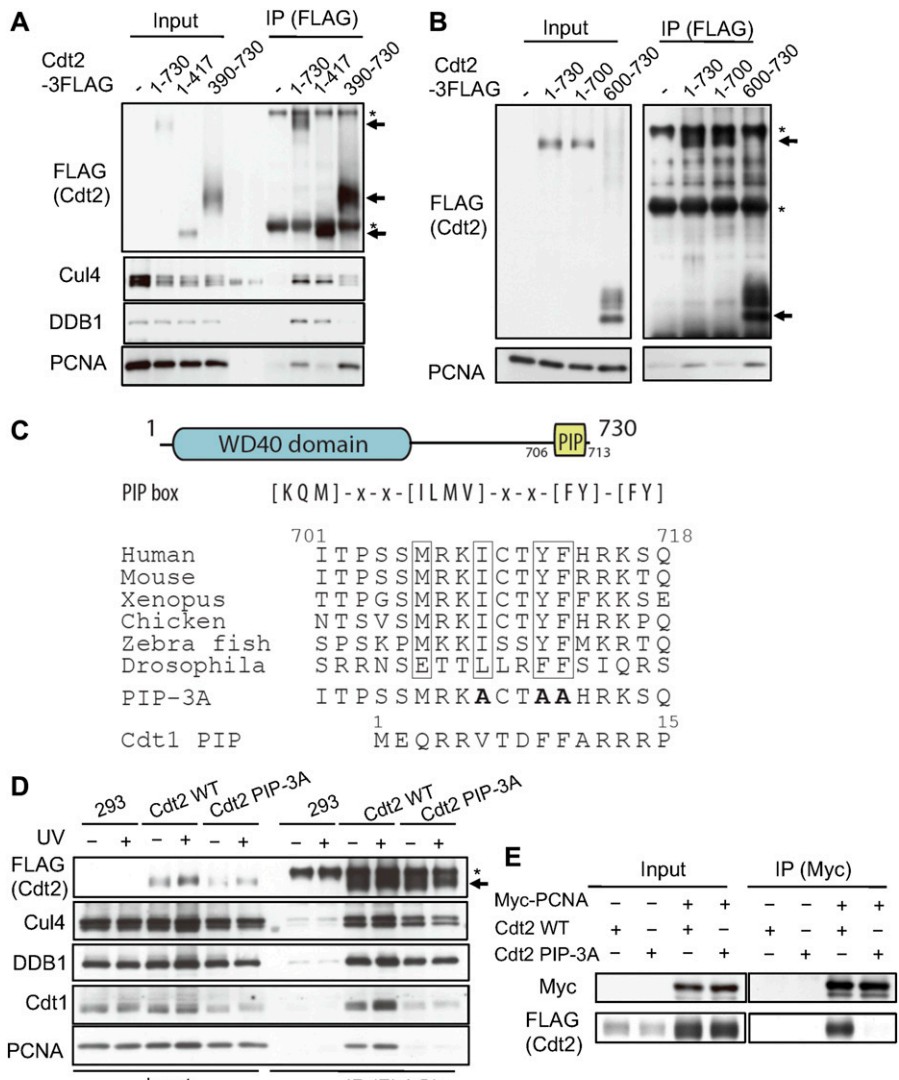

**Figure 3. The C-terminal domain of Cdt2 has a conserved PIP box and interacts with PCNA.**
**(A)** Full-length Cdt2 (Cdt2$^{1–730}$) and Cdt2$^{390–730}$, but not Cdt2$^{1–417}$, interact with PCNA. HEK293 cells were transfected with plasmids expressing the indicated Cdt2 fused to a C-terminal FLAG tag, cross-linked, and precipitated with anti-FLAG antibodies (arrows). Precipitates were examined for PCNA, Cul4A, and DDB1. * indicates nonspecific bands derived from IgGs. The bands in the blank lanes for Cul4A blotting correspond to the cross-reacting bands to pre-stained marker. **(B)** Cdt2$^{600–730}$, but not Cdt2$^{1–700}$, interacts with PCNA. HEK293 cells were transfected with the indicated plasmids and treated as in A. * indicates nonspecific bands derived from IgGs. **(C)** A schematic drawing of the Cdt2 domain structure highlighting the C-terminal PIP box and a sequence alignment of PIP boxes in various organisms. The alanine-changed PIP box mutant of Cdt2 was shown (PIP-3A) together with Cdt1 PIP box. **(D)** HEK293 cells were transfected with Cdt2$^{WT}$ or PIP box mutant (Cdt2$^{PIP-3A}$) C-terminally fused to a FLAG tag, treated as in B, and Cdt2 was precipitated with anti-FLAG resin. The interaction with PCNA and DDB1 was examined by immunoblotting. * indicates nonspecific bands. **(E)** HEK293 cells were transfected with Myc-tagged PCNA and Cdt2$^{WT}$-3FLAG or Cdt2$^{PIP-3A}$-3FLAG; PCNA was immunoprecipitated by anti-Myc antibodies and the interaction with Cdt2 was examined by immunoblotting.

To confirm the binding mode of the Cdt2$^{PIP}$ and Cdt1$^{PIP}$ peptides to PCNA, we determined the crystal structure of both complexes by X-ray crystallography, to a 3.5- and 3.4-Å resolution, respectively. PCNA adopts its well-characterized trimer conformation, with one peptide bound per monomer. The binding mode of both the peptides was very clear (Figs 4B, C, S3A and B and Table 1, see the Materials and Methods section for details). Both peptides bind in a similar manner to other PCNA complex structures. Some notable differences in the binding mode are the conserved Phe713/Phe11 side-chain ring that rotates 180 degrees between the two structures and the Tyr712/Phe10 that repositions so as to extend more towards a hydrophilic environment in Cdt2. The Ile709 and Val7 side chains occupy a similar space in the hydrophobic recognition pocket. Albeit the peptide main chain is in a very similar conformation between the structures, some of the other side chains adopt rather different conformations. The average buried area upon the binding of the Cdt2 peptide is 692 ± 6 Å$^2$ and average calculated energy of binding is −13 ± 0.2 kcal·mol$^{-1}$, whereas the average buried area

upon the binding of the Cdt1 peptide is 650 ± 4 Å$^2$ and average calculated energy of binding is −8.3 ± 0.5 kcal·mol$^{-1}$; these confirm the tighter binding of the Cdt2 peptide to PCNA.

## Cdt2 directly binds to PCNA on DNA through its PIP box independently of Cdt1 in vitro

The above results suggested that Cdt2 was recruited to the PCNA sites primarily through its PIP box rather than the N-terminal substrate receptor domain, which contrasts to the model that Cdt1 and CRL4$^{Cdt2}$ are sequentially recruited to PCNA$^{on Chromatin}$. To investigate the mechanism of Cdt1 and CRL4$^{Cdt2}$ recruitment to PCNA, we set up an in vitro analysis with purified and defined human proteins. FLAG-tagged Cdt1 (Cdt1-3FLAG) and CRL4$^{Cdt2}$ (Cdt2-3FLAG) were expressed in insect cells and purified on anti-FLAG resin (Fig 5A). PCNA and its loader replication factor C (RFC) were purified as described in the experimental procedures (Fig S4). First, we examined the interaction of Cdt1 or CRL4$^{Cdt2}$ with PCNA in the absence of DNA

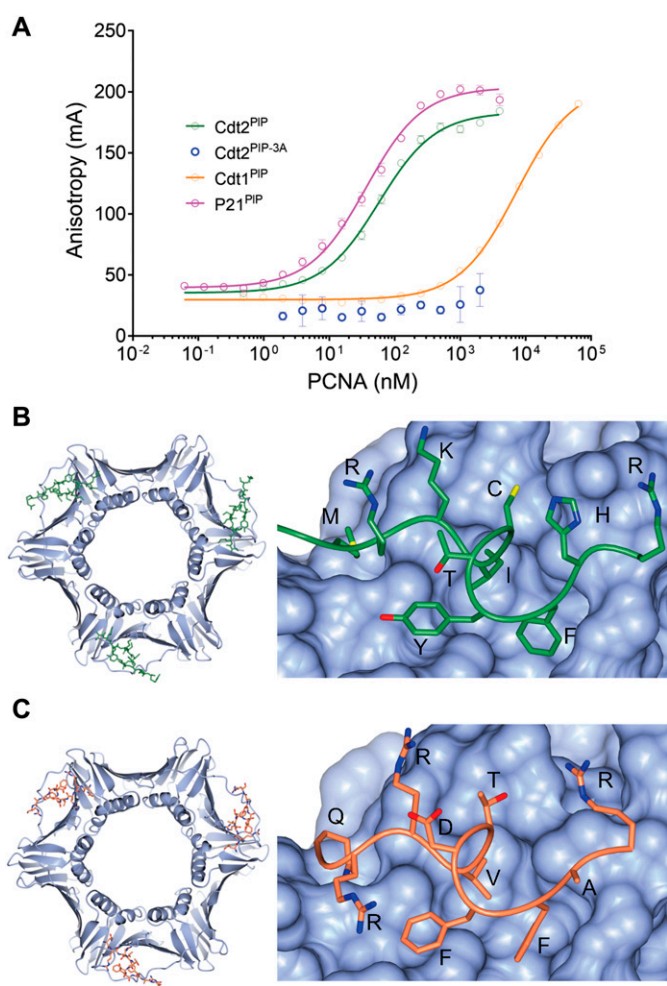

**Figure 4. Cdt2 PIP peptide directly interacts with PCNA.**
**(A)** An FP assay showing that the equilibrium-binding constants of the Cdt2 PIP box peptide (Cdt2^PIP), the triple mutant of key interacting residues of the classical PIP box (Cdt2^PIP-3A), as well as the Cdt1 PIP box peptide (Cdt1^PIP) and the p21 PIP box peptide (p21^PIP). **(B)** Crystal structure of the Cdt2^PIP peptide bound to PCNA (PDB code: 6QC0); both whole PCNA crystal structure image with Cdt2^PIP and zoomed-in image are shown. PCNA is shown as a blue surface; the Cdt2 peptide backbone is shown as a green "worm" model with side chains as cylinders; oxygen, red; nitrogen, blue; and sulfur, yellow. Amino acid residues are labelled. **(C)** Crystal structure of whole and zoomed-in images of the Cdt1^PIP bound to PCNA (PDB code: 6QCG); colors as before, but the Cdt1^PIP peptide is in orange.

(free PCNA). The purified Cdt1 or CRL4^Cdt2 were fixed on anti-FLAG beads, incubated with free PCNA, and the amount of bound PCNA was analysed. In contrast to the above-mentioned results that both Cdt1 and Cdt2 PIP box peptides bound to PCNA (Fig 4), PCNA was detected on Cdt1-beads, but much less on CRL4^Cdt2-beads (Fig 5B).

Because Cdt1 and Cdt2 bind to PCNA only when PCNA is loaded on DNA in *Xenopus* egg extracts (Havens & Walter, 2009), we prepared PCNA loaded on nicked circular DNA (ncDNA) with RFC (PCNA^on DNA) (Fig S5A and B). In our in vitro conditions, three molecules of PCNA trimer were loaded on one plasmid DNA. Then, we analysed the binding of Cdt1 and CRL4^Cdt2 to PCNA^on DNA as described in Fig S5A. After incubation, Cdt1 was efficiently recovered on the PCNA^on DNA beads, but not on the control beads (Fig 5C). Although we could not show CRL4^Cdt2 binding to free PCNA, we show that CRL4^Cdt2 binds to

PCNA^on DNA in the absence of Cdt1. The interaction was not mediated by RFC proteins bound on ncDNA (Fig 5D, lane 6). The molar ratio of Cdt2 bound to trimeric PCNA^on DNA was the same or somewhat higher than that of Cdt1 (408 fmol of Cdt2 bound to 158 fmol trimeric PCNA^on DNA, that is, 2.57 Cdt2 molecules on one PCNA trimer, while 301 fmol of Cdt1 to 148 fmol of trimeric PCNA^on DNA, 2.04 Cdt1 molecules on one PCNA trimer). These results indicate that CRL4^Cdt2 directly and efficiently binds to DNA-loaded PCNA without substrate, consistent with data in cells showing that Cdt2 was recruited to the sites of damage in the absence of Cdt1 (Fig 1).

To confirm that the interaction of CRL4^Cdt2 with PCNA^on DNA was mediated by the PIP box of Cdt2, the CRL4 complex having Cdt2^PIP-3A (CRL4^Cdt2(PIP-3A)) was purified like CRL4^Cdt2 (Fig 5A) and used in a binding assay. As above, CRL4^Cdt2 interacted with PCNA^on DNA. In contrast, the binding activity of CRL4^Cdt2(PIP-3A) to PCNA^on DNA was severely reduced as compared with CRL4^Cdt2 (Fig 5D). These results demonstrate that the PIP box of Cdt2 was directly responsible for the interaction of CRL4^Cdt2 with PCNA^on DNA.

### The Cdt2 PIP box promotes Cdt2 recruitment to UV-irradiated sites and PCNA foci

To investigate the role of the Cdt2 PIP box in cells, we isolated HEK293 cells stably expressing FLAG-tagged Cdt2^PIP-3A and showed

**Table 1. Data collection and refinement statistics.**

| PDB identifier | Cdt1/6QCG | Cdt2/6QC0 |
|---|---|---|
| Data Collection | | |
| Wavelength (Å) | 1.0000 | 0.8726 |
| Resolution (Å) | 38–3.4 (3.63–3.40) | 46–3.5 (3.83–3.50) |
| Space group | P $2_1$ $2_1$ $2_1$ | P $6_1$ |
| Unit cell a, b, c (Å) | 76.58, 143.73, 173.59 | 151.4, 151.4, 91.49 |
| $CC_{1/2}$ | 0.982 (0.647) | 0.976 (0.713) |
| $R_{merge}$ | 0.166 (0.631) | 0.289 (0.922) |
| $|I/\sigma I|$ | 7.6 (1.9) | 7.1 (2.1) |
| Completeness (%) | 99.3 (99.2) | 97.1 (88.1) |
| Multiplicity | 3.4 (3.5) | 8.0 (7.1) |
| Unique reflections | 26,761 | 14,830 |
| Refinement | | |
| PCNA/peptide in A.U. | 6 | 3 |
| Protein atoms | 24,564 | 12,324 |
| Averaged B-factors protein (Å$^2$) | 48.0 | 73.0 |
| $R_{work}/R_{free}$ (%) | 20.5/25.1 | 20.2/25.0 |
| Bond lengths rmsd/rmsZ (Å) | 0.0087/0.612 | 0.077/0.552 |
| Bond angles rmsd/rmsZ (°) | 1.682/0.981 | 1.709/0.997 |
| Ramachandran preferred/ outliers (%) | 89.1/1.9 | 89.1/1.8 |
| Clash score (%-ile) | 97 | 97 |
| MolProbity score (%-ile) | 92 | 90 |

High-resolution shell in parentheses.

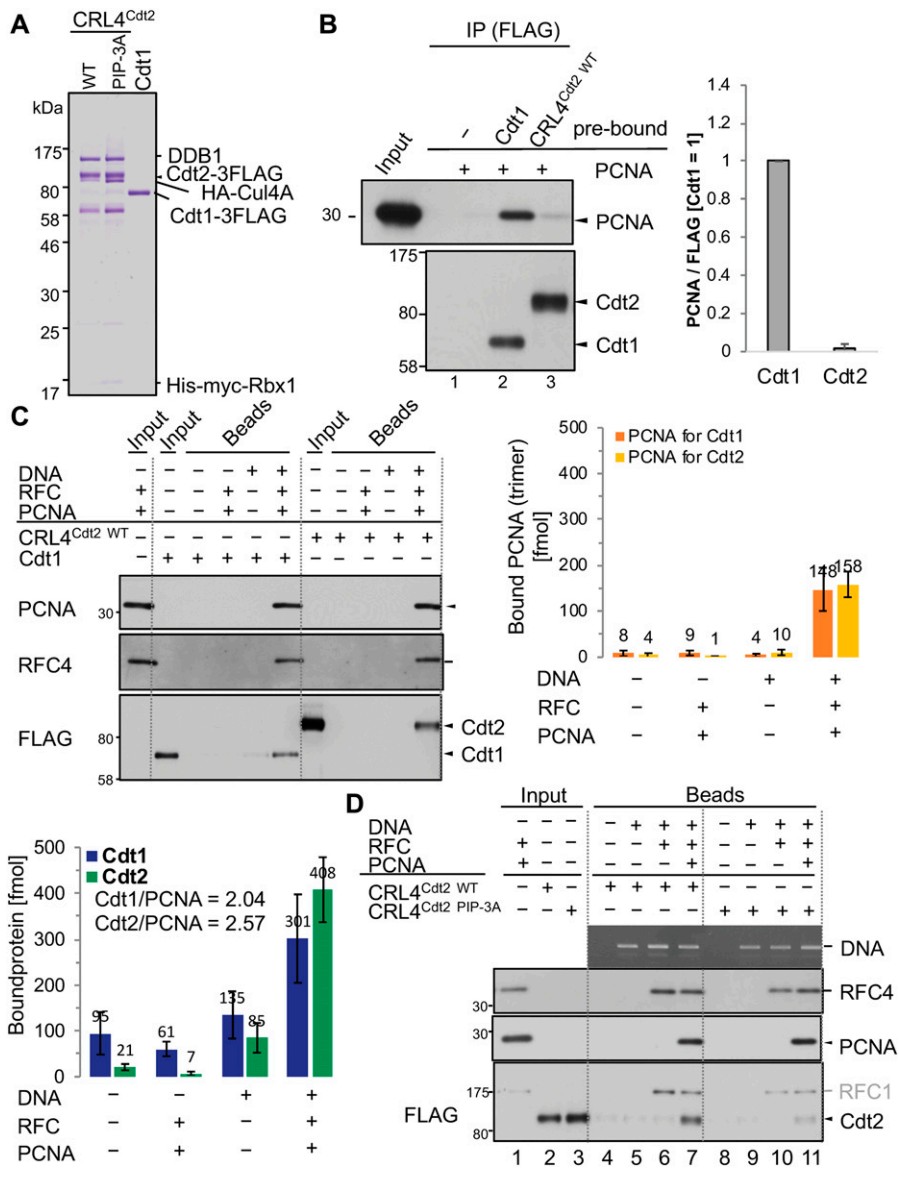

**Figure 5. Cdt1 and CRL4$^{Cdt2}$ independently associate with PCNA loaded on DNA through PIP box in vitro.**
**(A)** Purified CRL4$^{Cdt2(WT)}$, CRL4$^{Cdt2(PIP-3A)}$, and Cdt1 proteins. **(B)** Cdt1, but not CRL4$^{Cdt2}$, associates with free PCNA. The purified Cdt1 and CRL4$^{Cdt2}$ proteins were fixed on anti-FLAG–resins and incubated with PCNA for 2h at 4°C and the bound PCNA was analysed. The relative amounts of bound PCNA were shown, normalized with FLAG signal, and the level on Cdt1-3FLAG beads set to 1.0. Error bars represent SD from three independent experiments. **(C)** The ncDNA beads (DNA+) or control beads (DNA–) were loaded with PCNA by RFC or not. Then, beads were incubated with the purified Cdt1 or CRL4$^{Cdt2}$, and the bound proteins were recovered for Western blotting. The amounts of PCNA (as a trimer) loaded on plasmid DNA and the amounts of bound Cdt1-3FLAG and Cdt2-3FLAG were measured and shown (f mol). The number of bound Cdt1 and Cdt2 on PCNA trimer was shown (Cdt1/PCNA and Cdt2/PCNA). Error bars represent SD from independent experiments (n = 3 or n > 3). **(D)** CRL4$^{Cdt2(WT)}$ and CRL4$^{Cdt2(PIP-3A)}$ were incubated with control beads, ncDNA beads, or ncDNA beads which had been pre-incubated with RFC alone or together with PCNA for 1 h at 4°C. Beads were recovered and bound proteins were analysed.

that Cdt2$^{PIP-3A}$ was not recruited to UV-irradiated sites (Fig 6A). XPA, a component of nucleotide excision repair, staining was used to confirm that similar extents of DNA damage were induced both in Cdt2$^{WT}$- and Cdt2$^{PIP-3A}$-expressing cells. Next, we verified that the recruitment of Cdt2 onto PCNA during the S phase is also dependent on its PIP box. To detect the chromatin-associated fraction of PCNA and Cdt2, asynchronous cells were pre-extracted before fixation and co-stained with PCNA and FLAG antibodies. A similar percentage of PCNA-positive S-phase cells was detected both in Cdt2$^{1–730/WT}$-expressing cells and Cdt2$^{PIP-3A}$-expressing cells (Fig S6). Cdt2$^{WT}$ was co-localized with PCNA foci, corresponding to sites of DNA replication. More than 80% of PCNA-positive cells were co-stained with Cdt2$^{WT}$, with the early S-phase cells displaying stronger Cdt2$^{WT}$ staining than the late S-phase cells as reported (Nukina et al, 2018) (Figs 6B and C, and S6A). In contrast, almost no signal was detected for Cdt2$^{PIP-3A}$ irrespective of the presence of

PCNA. Consistent with the immunofluorescence analysis, following cell fractionation, Cdt2$^{WT}$ was recovered in a chromatin-containing fraction, whereas Cdt2$^{PIP-3A}$ was undetectable (Fig 6D).

These results collectively suggest that although Cdt2$^{PIP-3A}$ is capable of forming a complex with the rest of the components of the CRL4 ligase, the Cdt2 PIP box is required for efficient recruitment of CRL4$^{Cdt2}$ to the PCNA-loaded sites.

## Cdt1 ubiquitination is abortive in Cdt2$^{PIP-3A}$-expressing cells

Because the recruitment of Cdt2$^{PIP-3A}$ to PCNA sites was compromised, it would imply that ubiquitination of Cdt1 was also abrogated. To consider this possibility, asynchronously growing control HEK293 cells and cells expressing Cdt2$^{WT}$ or Cdt2$^{PIP-3A}$ were treated with proteasome inhibitor MG132 and the levels of poly-ubiquitinated Cdt1 were examined. Although poly-ubiquitinated

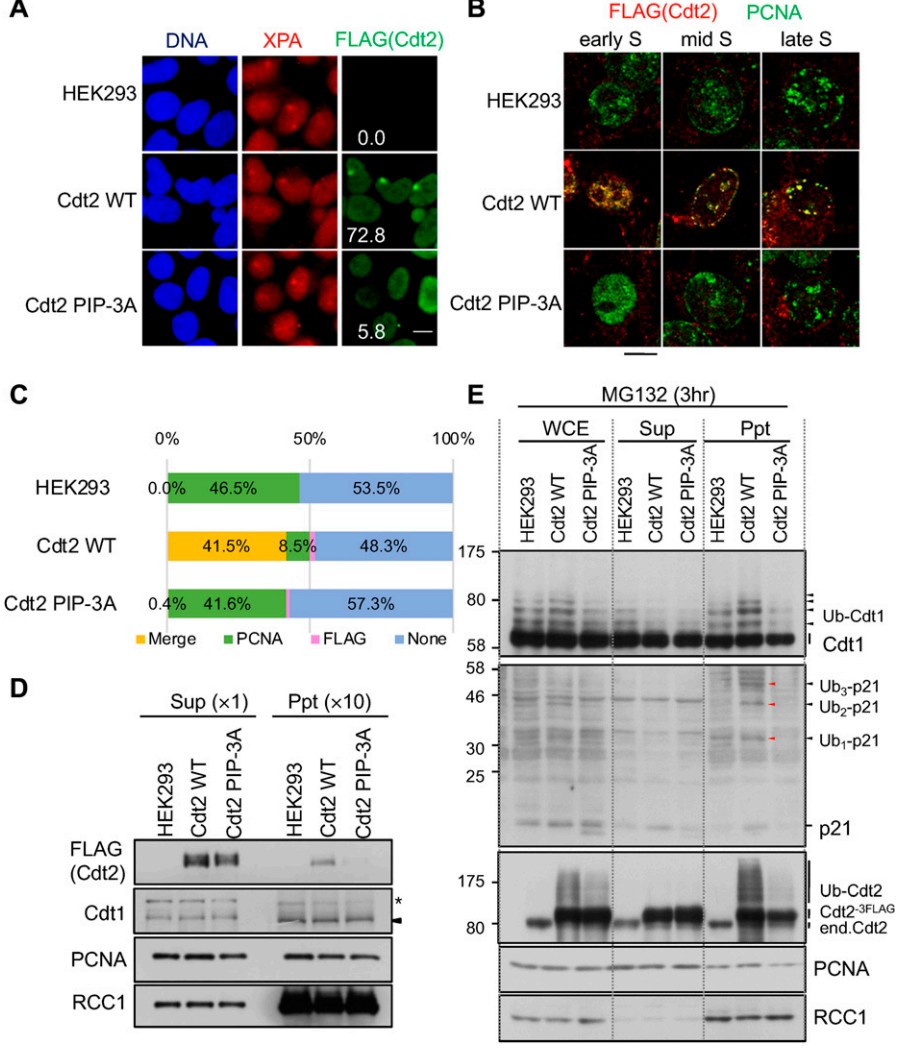

**Figure 6. The Cdt2 PIP box is important for its recruitment to DNA damage sites and replication foci and for poly-ubiquitination activity.**
**(A)** HEK293 and stable Cdt2$^{WT}$ or Cdt2$^{PIP-3A}$ cells were locally UV irradiated, and stained for XPA and FLAG (Cdt2). XPA-positive spots were examined for FLAG signal, and its frequency was counted (%). More than 50 spots were examined. bar:10 $\mu$m. **(B)** HEK293 cells and stable cells were pre-extracted, fixed, and stained with PCNA and DYKDDDDK (FLAG) antibodies. Bar: 10 $\mu$m. **(C)** Cells stained as (B) were examined for staining with PCNA and DYKDDDDK (FLAG) antibodies and the frequency of cells with colocalized PCNA and FLAG (Cdt2) staining (Merge), PCNA staining alone (PCNA), FLAG staining alone (FLAG), or no staining (None) were plotted. **(D)** Whole-cell extracts were prepared from HEK293 and stably transfected cells of Cdt2$^{WT}$ or Cdt2$^{PIP-3A}$ cells, separated into soluble (Sup) and chromatin-containing insoluble (Ppt) fractions, and were examined for indicated proteins. The asterisk indicates nonspecific bands. **(E)** HEK293 and stably transfected cells of Cdt2$^{WT}$ or Cdt2$^{PIP-3A}$ were treated with MG132 for 3 h. Whole-cell extracts (WCE) were prepared, separated into soluble (Sup) and insoluble (Ppt) fractions, and blotted with the indicated antibodies.

Cdt1 was clearly detected in Cdt2$^{WT}$-expressing cells, its levels were reduced in Cdt2$^{PIP-3A}$-expressing cells (Fig 6E), indicating that the ability of Cdt2$^{PIP-3A}$ to ubiquitinate Cdt1 was compromised. Fractionation of cell extracts revealed that most of the poly-ubiquitinated Cdt1 and p21 were recovered in the chromatin-containing precipitate fraction prepared from Cdt2$^{WT}$-expressing cells, in agreement with CRL4$^{Cdt2}$ operating on chromatin (Fig 6E). Cdt2$^{PIP-3A}$-expressing cells however had reduced level of both poly-ubiquitinated Cdt1 and p21 on chromatin. Interestingly, Cdt2 itself is poly-ubiquitinated on chromatin, and this is severely reduced in Cdt2$^{PIP-3A}$-expressing cells (Fig 6E). Increased levels of poly-ubiquitination of Cdt1 and p21 were also observed in another cell line, U2OS cells expressing Cdt2$^{WT}$, but not in cells expressing Cdt2$^{PIP-3A}$ (Fig S7B).

Next, we examined the accumulation of poly-ubiquitinated Cdt1 after UV irradiation. For this experiment, the endogenous Cdt2 was depleted with an siRNA targeted to the 3′ UTR to assay for the activity of introduced Cdt2. Cells were treated with MG132, irradiated with UV and the levels of poly-ubiquitinated Cdt1 were monitored by Western blotting. In HEK293 cells treated with

control siRNA, poly-ubiquitinated Cdt1 was detected 15 min after irradiation (Fig S6B). Depletion of Cdt2 blocked the poly-ubiquitination of Cdt1. The defect was rescued by Cdt2$^{WT}$ expression. In contrast, Cdt2$^{PIP-3A}$-expressing cells were defective in poly-ubiquitination of Cdt1 (Fig S6B), suggesting that Cdt2 PIP box is required for efficient poly-ubiquitination of substrates.

### The Cdt2 PIP box is required for efficient Cdt1 degradation

Because the poly-ubiquitination activity of Cdt2$^{PIP-3A}$ was reduced, we monitored the degradation of Cdt1 after UV irradiation. Asynchronously growing control U2OS cells and cells expressing Cdt2$^{WT}$ or Cdt2$^{PIP-3A}$ (Fig S7) were depleted of endogenous Cdt2 using an siRNA targeted to the 3′ UTR, and irradiated with UV. When the cells were exposed to UV (20 J/m$^2$) radiation, degradation of Cdt1 was almost prevented in U2OS cells depleted of endogenous Cdt2 (Fig 7A and B). Ectopic expression of Cdt2$^{WT}$ restored Cdt1 degradation to similar kinetics to that of control siRNA-transfected U2OS cells. In contrast, Cdt2$^{PIP-3A}$ was much less effective in rescuing Cdt2 depletion. Similarly, at a lower dose of UV irradiation (5 J/m$^2$), Cdt1

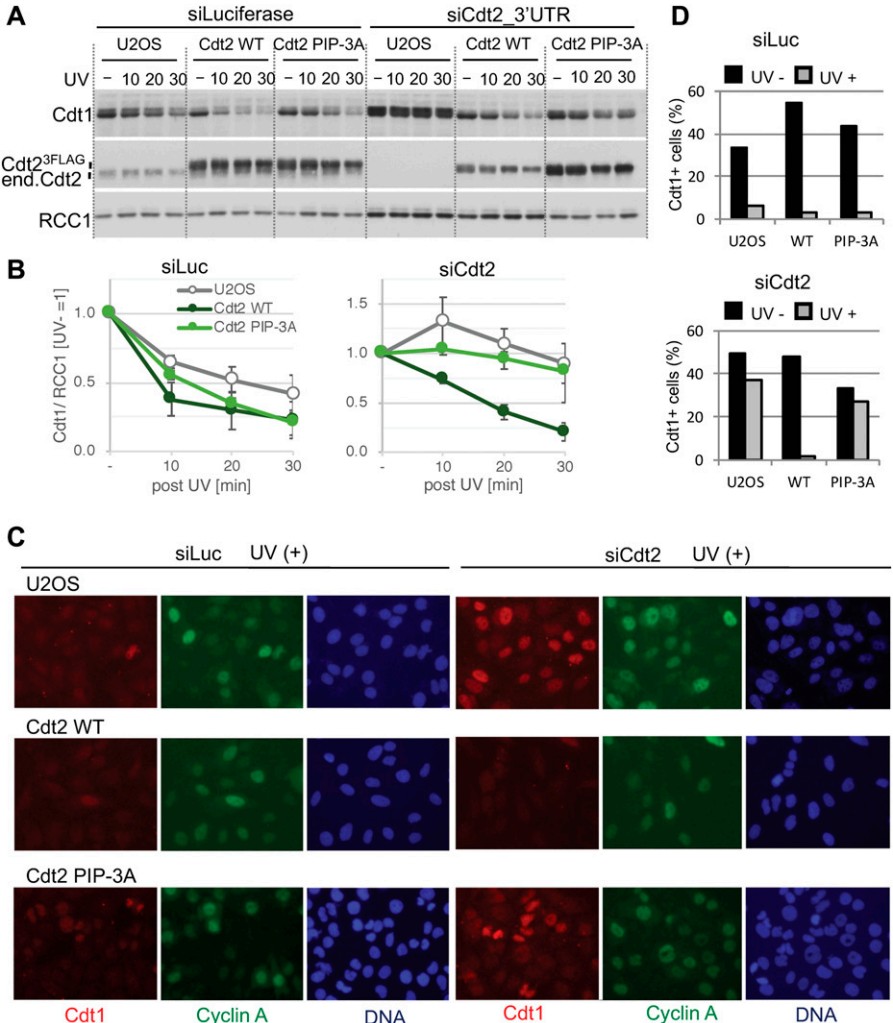

**Figure 7. The Cdt2 PIP box plays an important role for rapid Cdt1 degradation after UV irradiation.**
**(A)** U2OS cells and U2OS cells stably expressing Cdt2[WT]-3FLAG or Cdt2[PIP-3A]-3FLAG were transfected with siCdt2 targeted to the 3′ UTR or control siLuc for 48 h, irradiated with UV (+, 20 J/m²) or not (−), and collected at the indicated time points (min) for Western blotting. **(B)** The protein levels were measured and the relative amounts of Cdt1, normalized to RCC1, after UV irradiation were shown, UV (−) level set as 1.0. Error bars represent SD from three independent experiments. **(C)** Cells were transfected with siCdt2 or control siLuc for 72 h, irradiated with UV (+, 5 J/m²) or not (−) (not shown), and 1h later fixed for staining with Cdt1 and cyclinA antibodies. **(D)** The frequency of Cdt1-positive (+) cells among the total cells in (C) was shown (%) (average of two independent experiments).

remained stable 1 h post UV irradiation in Cdt2[PIP-3A] cells, when endogenous Cdt2 was depleted, both on immunofluorescent analyses and Western blotting (Figs 7C and D, and S7C).

Next, we examined Cdt1 degradation after the onset of the S phase in the Cdt2[PIP-3A]-expressing cells. Cyclin A is present in the S and G2 phase, when Cdt1 should be degraded. Thus, Cdt1 should be present in cells that are negative for cyclin A, and absent in cells that are positive for cyclin A (Nishitani et al, 2006; Xouri et al., 2007a, 2007b). Cdt1 degradation after the onset of the S phase is carried out by two ubiquitin ligases, CRL1[Skp2] and CRL4[Cdt2]. We depleted the substrate recognition subunits, Cdt2 and/or Skp2, by siRNA, and the percentage of Cdt1 positive cells among cyclin A–positive cells was assessed by immunofluorescence. In control siRNA-transfected U2OS cells, ~2% of cyclin A–positive cells stained positive for Cdt1. In contrast, in Cdt2-depleted, Skp2-depleted, and Cdt2/Skp2 co-depleted cells, 50%, 30%, and more than 80% of cyclin A–positive cells stained positive for Cdt1, respectively (Fig 8A–C), consistent with earlier work (Nishitani et al, 2006). Then, we depleted Cdt2 and Skp2 and examined Cdt1 degradation in Cdt2[WT] or Cdt2[PIP-3A]-expressing cells. Ectopic expression of Cdt2[WT] fully complemented the depletion

of Cdt2, and also the co-depletion of Skp2. This was probably because the levels of ectopically expressed Cdt2[WT] were higher than the endogenous Cdt2 levels. On the other hand, Cdt1 degradation was not fully rescued by the expression of Cdt2[PIP-3A] protein both in Cdt2-depleted and Cdt2 and Skp2 co-depleted cells (Fig 8A and B). Upon depletion of endogenous Cdt2, many cells exhibited significantly enlarged nuclei, consistent with re-replication. This phenotype was rescued by ectopic expression of Cdt2[WT], but not of Cdt2[PIP-3A] (Figs 8D and S8), suggesting that the PIP box of Cdt2 is essential to prevent re-replication.

Taken together, our in vitro and in vivo analyses show that the Cdt2 PIP box brings CRL4 ubiquitin ligase to PCNA sites and is required for efficient substrate degradation both during the S phase and following UV irradiation, to maintain genome stability.

## Discussion

Cdt2 substrates are degraded rapidly as cells enter the S phase or when cells are exposed to DNA-damaging agents. Cdt2 of higher

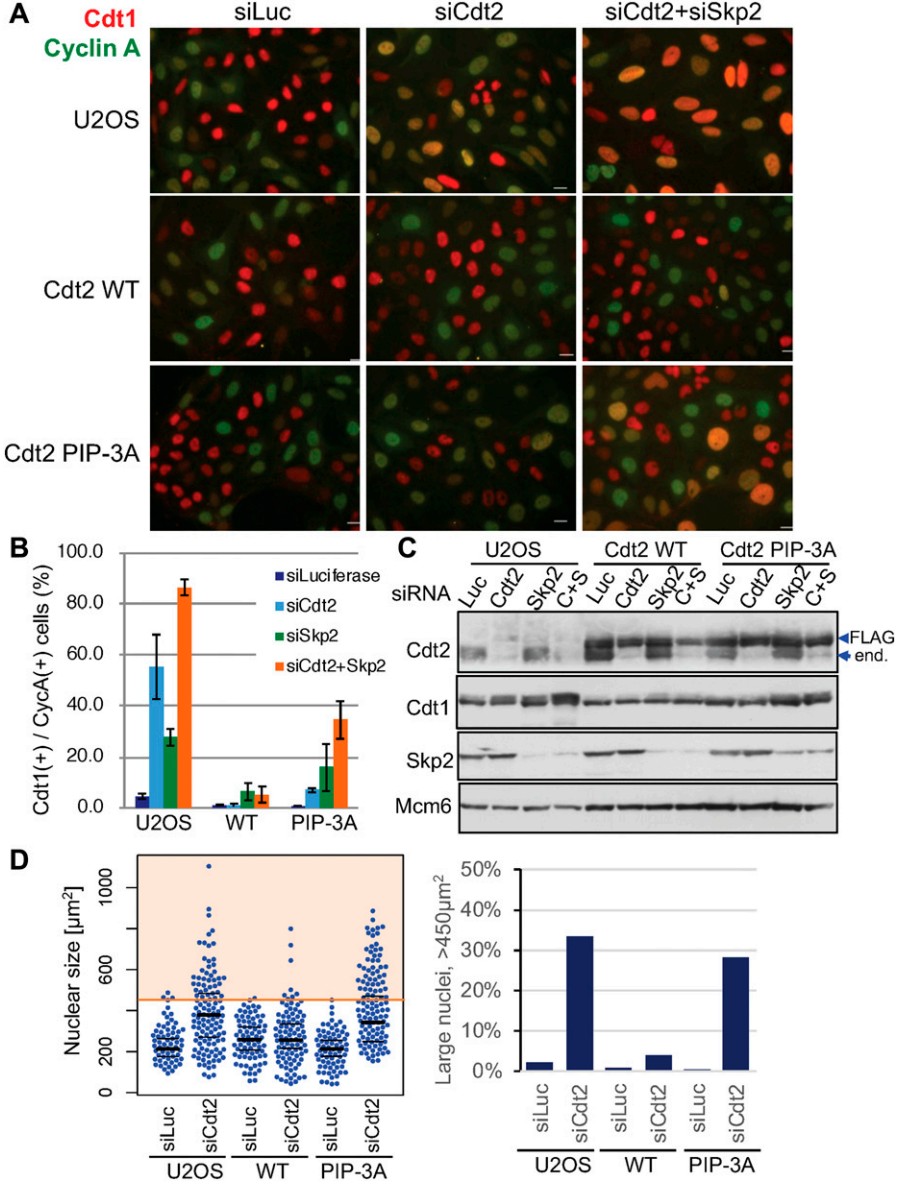

**Figure 8. The Cdt2 PIP box plays an important role for Cdt1 degradation during cell cycle.**
(**A, B**) U2OS cells and U2OS cells stably expressing Cdt2^WT-3FLAG or Cdt2^PIP-3A-3FLAG were transfected with siCdt2 and/or siSkp2 or control siLuc for 72 h. Cells were fixed and stained with Cdt1 and cyclinA antibodies. The frequency of Cdt1-positive (+) cells among cyclin A–positive (+) cells was calculated (n = 3). (**C**) Cells transfected with siRNAs as above (C+S; siRNAs for Cdt2 and Skp2) were examined for indicated protein levels. (**D**) Cells treated with siCdt2 or control siLuc were measured for their nuclear size (n = 200 for each), and are shown in a scatter plot. Frequency of cells with nuclear size larger than 450 $\mu m^2$ is shown (%).

eukaryotic cells have an extended C-terminal region. We demonstrate that Cdt2 has a PIP box in its C-terminal end that recognizes chromatin-bound PCNA with high affinity. This interaction is important for CRL4^Cdt2 function in recognizing its targets before their ubiquitin-dependent degradation: this previously unnoticed Cdt2 PIP box is required for recruitment of CRL4^Cdt2 to chromatin-bound PCNA, and indispensable for efficient Cdt1 ubiquitination, thus for prompt degradation of Cdt1 both in the S phase and after UV irradiation. The Cdt2 PIP box peptide directly interacts with PCNA with high affinity (about 50 nM) in vitro, similar to the unusually high-affinity p21 PIP box peptide, and significantly tighter than the Cdt1 PIP box peptide. Molecular modelling experiments suggest that both the Cdt1 and Cdt2 PIP boxes bind PCNA in the classical way and their binding to PCNA occurs independently of each other (Fig 4). Furthermore, the purified CRL4^Cdt2 ligase interacted directly with

the PCNA loaded on DNA via Cdt2 PIP box (Fig 5). Consistently, the C-terminal domain of Cdt2 has been shown to bind to PCNA in *Xenopus* egg extracts (Kim et al, 2010). Our analysis now shows that the PIP box is conserved in *Xenopus*, and thus likely to be responsible for this interaction.

PCNA is a trimer, hence it offers three PIP box binding sites that are proposed to be asymmetric when PCNA is loaded onto DNA (Ivanov et al, 2006). Consequently, both Cdt1 and Cdt2 can be simultaneously recruited onto PCNA, effectively utilizing PCNA as a common docking platform for bringing the CRL4^Cdt2 ligase and the Cdt1 substrate in proximity, allowing more efficient transfer of a ubiquitin moiety from the E2 to Cdt1 (Fig 9). This Cdt2 PIP box assisted ubiquitination is likely to be used by other Cdt2 substrates, as p21 poly-ubiquitination was also defective in Cdt2^PIP-3A-expressing cells (Figs 6E and S7B). This mechanism, where co-localization onto the

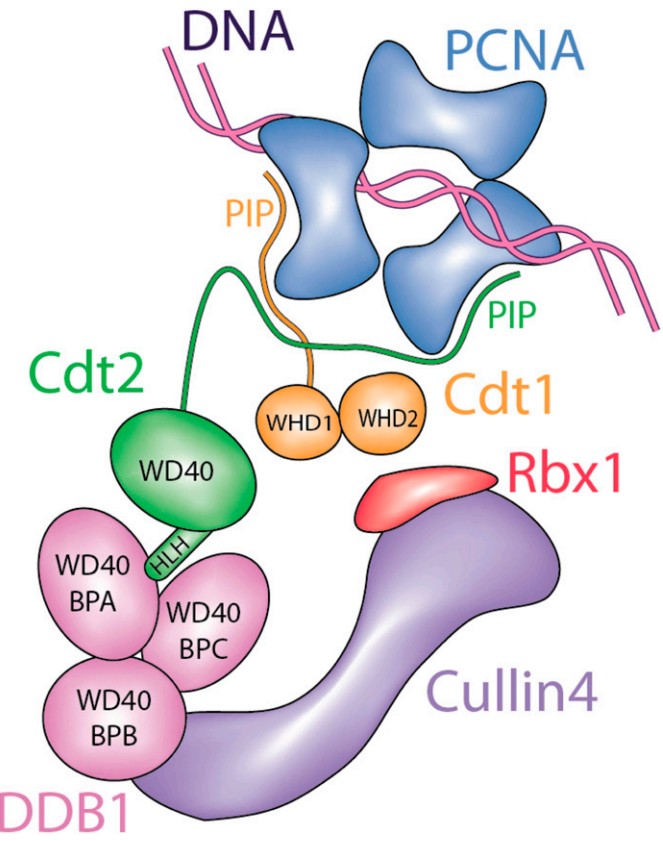

**Figure 9. Model.**
A model showing how both CRL4[Cdt2] and Cdt1 are likely recruited to the chromatin-bound PCNA through their PIP boxes, for efficient ubiquitination.

same protein platform brings the E3 ubiquitin ligase and its substrate in proximity, is a new paradigm in ubiquitin conjugation. This model fits recent results which show that Cdt2 itself can be auto-ubiquitinated by CRL4[Cdt2] (Abbas et al, 2013): CRL4[Cdt2] and Cdt2 (or a second CRL4[Cdt2]) may both bind PCNA through the Cdt2 PIP box and catalyse this auto-ubiquitination event. Consistently, the poly-ubiquitination of Cdt2 which was observed on chromatin in Cdt2[WT]-expressing HEK293 cells was severely reduced in Cdt2[PIP-3A]-expressing cells (Fig 6E).

The mono-ubiquitination (mUb) of PCNA is carried out by Rad6-Rad18, but CRL4[Cdt2] also contributes to it. In non-damaged cells, mUb of PCNA is present at a basal level and its levels increase in response to DNA damage (Terai et al., 2013). The Cdt2 PIP box may also be important for mUb of PCNA, as mUb-PCNA levels appeared decreased in Cdt2[PIP-3A] in comparison with Cdt2[wt]-expressing cells (Fig S7B). Therefore, Cdt2 PIP box may be required for various events driven by CRL4[Cdt2] in the cell; degradation of PIP-degron proteins, auto-ubiquitination of Cdt2, and PCNA mUb.

We show that the Cdt2 PIP box is crucial for recruitment of the CRL4[Cdt2] ligase onto PCNA in human cells. However, further interactions of Cdt2 domains, mediated by its N-terminal WD40 repeats, with the PCNA-bound Cdt1 PIP box (the PIP degron) also enhance the affinity of the CRL4[Cdt2] (Havens & Walter, 2009). Here, we propose a synergistic mechanism where the Cdt2 PIP box and the Cdt1 PIP-degron operate in concert to promote

CRL4[Cdt2]-dependent target ubiquitination (Figs 9 and S9). It is also noteworthy that the binding of Cdt1 onto PCNA is transient in cells, whereas Cdt2 interactions are more stable (Roukos et al, 2011 and Fig 1B): a synergistic mechanism would imply that interactions of pre-bound Cdt2 with transiently formed Cdt1 PIP-degron, are important for localizing CRL4[Cdt2] and Cdt1 onto PCNA. The property that affinity of Cdt2 PIP box to PCNA is higher than that of Cdt1 PIP box (Fig 4A) is compatible with such a model. In addition, weaker affinity of Cdt1 PIP box can help to release Cdt1 after poly-ubiquitination, whereas CRL4[Cdt2] could remain on PCNA to trap the next substrate, leading to an efficient degradation cycle of substrates (Fig S9). In the absence of Cdt2 PIP box, the process needs to be performed sequentially; transient Cdt1 binding to PCNA and recognition by Cdt2 via its N-terminal WD40 repeats, which would be, however, less effective for Cdt1 degradation.

Our biochemical analysis demonstrated that CRL4[Cdt2] interacts more strongly with PCNA[on DNA] than with PCNA that is not associated with DNA (free PCNA) (Fig 5B and C). Because Cdt2 PIP peptide can bind to free PCNA (Fig 4), there is a mechanism that enhances the affinity of CRL4[Cdt2] to the PCNA[on DNA]. This is compatible with models that suggest that a conformational change upon DNA binding can modulate affinity. It is tempting to speculate that the high-affinity CRL4[Cdt2] complex docking into DNA-bound PCNA triggers changes that allow the transient loading of low-affinity substrates such as Cdt1, enabling rapid ubiquitination molecules in the vicinity. On the other hand, we reproducibly observed that Cdt2 was recovered, although at lower levels, on DNA beads in the absence of PCNA (Fig 5C), suggesting that CRL4[Cdt2] has a DNA-binding activity. Furthermore, the RFC1 complex may have a role connecting PCNA and Cdt2, as Cdt2 PIP-3A was detected to some more levels in the presence of the PCNA and RFC1 complex (Fig 5D). As reported, PCNA loaders appear to have an additional role in the CRL4[Cdt2]-mediated ubiquitination after loading PCNA (Shiomi et al, 2012). The extended C-terminal region of Cdt2 might have an additional domain involved in such a regulation, or an unknown factor might be involved. The exact molecular mechanisms that interplay to regulate these interactions need to be investigated further.

Although the Cdt2 PIP box mediates direct interaction with PCNA[on DNA], the affinity of Cdt2 to PCNA decreases as cells progress into and complete the S phase. The chromatin levels of Cdt2 are reduced during the late S phase (Rizzardi et al, 2015), and the co-localization of Cdt2[WT] with PCNA was mostly observed in the early S phase cell nuclei (Fig 6B). It was suggested that the CDK-dependent phosphorylation of Cdt2 down-regulates Cdt2 interaction with PCNA, as Cdk1 inhibition recovered the chromatin interaction of Cdt2 at the late S phase, and Cdt2 mutated at CDK phosphorylation sites displayed strong interaction with PCNA throughout the S phase (Rizzardi et al, 2015; Nukina et al, 2018). We propose that the C-terminal region of Cdt2 contributes to regulating the activity of CRL4[Cdt2] in the cell cycle by controlling its affinity to PCNA; Cdt2 PIP box–mediated binding to PCNA is important for rapid substrate degradation at the onset of the S phase, whereas the phosphorylation of the C-terminal region of Cdt2 inhibits its PCNA interaction and thus reduces ubiquitination activity during the late S phase, leading to substrate re-accumulation. It remains, however, to be elucidated how phosphorylation of Cdt2 affects its interaction to PCNA. We noticed that our Cdt2 preparation from insect cells was phosphorylated (Fig S4D)

at levels found in human cells. The assay with de-phosphorylated Cdt2 and kinase-treated Cdt2 could help to understand how the phosphorylation on Cdt2 regulate its binding activity.

CRL-type ligases are thought to use the "SRF" or "substrate receptor" subunit to recognize a specific degron in the substrate itself. However, CRL4-type ligases can deviate from that mechanism. For example, CRL4$^{DDB2}$ recognizes specific UV-damaged bases in DNA, creating a "ubiquitination zone" around the repair site, to ubiquitinate specific substrates (Fischer et al, 2011). Here, we show a novel specific mechanism, where the ligase and substrate recognize the same molecule of a cellular platform, to bring the enzyme and substrate into close proximity and thus to facilitate substrate trapping: our data constitute a new paradigm for the use of a docking platform in ubiquitin conjugation. Creating such a three-component system offers many possibilities for regulation and robustness of the ubiquitination response: the recognition of PCNA by both substrate and Cdt2 through specialized PIP boxes likely enables further interactions between substrate and Cdt2 (as, for example, the recognition of the PIP degron by its N-terminal WD40 repeat domain), creating a set of redundant molecular recognition events that regulate this system in the cellular context.

Recently published work is consistent with and supports our data (Leng et al, 2018).

## Experimental Procedures

### Cell culture

HeLa cells, HEK293 cells, 293T cells, and U2OS cells were cultured in Dulbecco's modified Eagle's medium with 10% fetal bovine serum and 5% $CO_2$. MCF7 cells were cultured in Dulbecco's modified Eagle's medium with 20% fetal bovine serum and 5% $CO_2$ at 37°C. HEK293 cells and U2OS cells stably expressing Cdt2$^{WT(1-730)}$-3FLAG, Cdt2$^{1-417}$-3FLAG, FLAG-Cdt2$^{390-730}$, or Cdt2$^{PIP-3A}$-FLAG were isolated using plasmids pCMV-HA-Cdt2$^{WT(1-730)}$-3FLAG, pCMV-HA-Cdt2$^{1-417}$-3FLAG, p3FLAG-3NLS-myc-Cdt2$^{390-730}$, or pCMV-HA-Cdt2$^{PIP-3A}$-3FLAG. Proteasome inhibitor MG132 was used at 25 $\mu M$. UV-C (254 nm) irradiation of whole cells in dishes was performed at 5 to 100 J/m$^2$ using a UV lamp (SUV-16, As One, Japan), a UV cross-linker (FS-800, Funakoshi) or a UV box equipped with a UV-C lamp and a radiometer 254 nm.

### Plasmids

The Cdt2$^{WT(1-730)}$-3FLAG–expressing plasmid, pCMV-HA-Cdt2$^{WT(1-730)}$-3FLAG was constructed by cloning the PCR-amplified HA-Cdt2$^{WT(1-730)}$ fragment into NcoI-NotI-cut pCMV-3FLAG. The Cdt2$^{1-417}$-3FLAG–expressing plasmid, pCMV-HA-Cdt2$^{1-417}$-3FLAG, and pCMV-HA-Cdt2$^{1-700}$-3FLAG were constructed in the same way by PCR-amplifying the corresponding region. FLAG-Cdt2$^{390-730}$, and Cdt2$^{600-730}$-expressing plasmids were constructed by PCR-amplifying the corresponding regions, (390–730, and 600–730) and cloning into the p3FLAG-3NLS-myc plasmid which was prepared by cutting out the p21 gene from the p3FLAG-3NLS-myc-p21 plasmid (Nishitani et al, 2008). pcDNA-6myc-PCNA was constructed by ligating the PCR-amplified 6myc and PCNA. To construct the expression

plasmid of PIP box mutant of Cdt2 (Cdt2$^{PIP-3A}$-FLAG), pCMV-HA-Cdt2$^{PIP-3A}$-3FLAG, Quick Change site-directed mutagenesis method (StrateGene) was performed using primers: AGCTCCATGAGGAAA-GCCTGCACAGCCGCCCATAGAAAGTCCCAGGAG and CTGGGACTTTCTAT-GGGCGGCTGTGCAGGCTTTCCTCATGGAGCTGGG. The baculovirus expression plasmids for HA-Cul4A and His-myc-Rbx1 were as described (Nishitani et al, 2008). The pBP8-Cdt2$^{WT(1-730)}$-3FLAG plasmid was constructed by ligating the BamH1-Cdt2-Kpn1 fragment from pBP8-Cdt2 and the Kpn1-Cdt2-3FLAG-Sma1 fragment from pCMV-HA-Cdt2-3FLAG into pBacPAK8 between BamH1 and Sma1 sites. pBP9-DDB1 was constructed by ligating the PCR-amplified BamH1-DDB1-EcoR1 fragment and the EcoR1-DDB1-Xho1 fragment into pBacPAK9 at BamH1-Xho1 sites. pBP9-Cdt1-3FLAG was constructed by cloning the PCR-amplified Sac1-Cdt1-3FLAG-Xho1 fragment using pCMV-Cdt1-3FLAG as a template into pBacPAK9. The GFP-Cdt2$^{wt}$, GFP-Cdt2$^{(1-417)}$, and GFP-Cdt2$^{(390-730)}$ expression plasmids were constructed by subcloning of the respective FLAG-tagged constructs into pEGFP-C3 (Clontech). For the GFP-Cdt2$^{wt}$–expressing plasmid, pCMV-HA-Cdt2$^{WT(1-730)}$-3FLAG was cut with NcoI and KpnI, KpnI and XmaI. To produce blunt ends for the NcoI sites, Klenow was used. The fragment was then cloned into the HindIII and XmaI sites of pEGFP-C3. The ends produced by HindIII were made blunt by Klenow. For the GFP-Cdt2$^{(1-417)}$, the pCMV-HA-Cdt2$^{1-417}$-3FLAG construct was cut by NcoI and XmaI. Klenow was used for the NcoI ends. The fragment was then cloned into the HindIII and XmaI sites of pEGFP-C3. The ends produced by HindIII were made blunt by Klenow. To achieve a better nuclear localization for the GFP-Cdt2$^{wt}$ and GFP-Cdt2$^{(1-417)}$ constructs, these were subcloned into p3FLAG-3NLS-myc by XhoI and BamHI. Then, they were subcloned again into pEGFP-C3 by using the HindIII-BamHI sites. For the GFP-Cdt2$^{(390-730)}$ expression plasmid, the p3FLAG-3NLS-myc Cdt2$^{(390-730)}$ construct was cut with HindIII and BamHI. The fragment produced was cloned into the HindIII-BamHI sites of pEGFP-C3.

### Antibodies, Western blotting, and immunofluorescence

For Western blotting, whole-cell lysates were prepared by lysing cell pellets directly in SDS–PAGE buffer. For immunofluorescence, U2OS or HEK293 cells were fixed in 4% paraformaldehyde (PFA; WAKO) for 10 min, permeabilized in 0.25% (vol/vol) Triton X-100 in PBS, and stained with the indicated antibodies as described previously. For double-staining with PCNA and Cdt2$^{WT}$ or Cdt2$^{PIP-3A}$, cells were permeabilized in 0.1% Triton X-100 for 1 min on ice, and were fixed in 4% PFA for 10 min at room temperature, followed by fixation in ice-cold methanol for 10 min. For cyclobutane pyrimidine dimer (CPDs) staining, MCF7 and HeLa cells were fixed in 4% PFA for 10 min, permeabilized in 0.3% Triton X-100 in PBS, and then incubated in 0.5 N NaOH for 5 min for DNA denaturation before immunofluorescence. Alexa488-conjugated anti-mouse and Alexa592-conjugated anti-rabbit antibodies were used as secondary antibodies with Hoechst 33258 to visualize DNA. The following primary antibodies were used: Cdt1 (Nishitani et al, 2006), Cdt2 (Nishitani et al, 2008), cyclin A (mouse, Ab-6; Neomarkers; rabbit, H-432; Santa Cruz), Myc (mouse, 9E10; Santa Cruz), FLAG (F3165, F7425; Sigma-Aldrich), PCNA (PC10; Santa Cruz; rabbit serum, a gift from Dr. Tsurimoto), XP-A (FL-273; Santa Cruz), RCC1 (Shiomi et al, 2012), DDB1 (Bethyl Laboratories), Rbx1 (ROC1, Ab-1;

Neomarkers), Cul4A (Bethyl Laboratories), RFC4 (H-183; Santa Cruz Biotechnology), DYKDDDDK Tag (Cell Signaling), and CPDs (CosmoBio). Protein levels were analysed by ImageJ software.

## Micropore UV irradiation assay

We used a method to induce DNA photoproducts within localized areas of the cell nucleus as described in Ishii et al (2010). To perform micropore UV irradiation, cells were cultured on cover slips, washed twice with PBS, and subsequently covered with an isopore polycarbonate membrane filter (Millipore) with a pore size of either 3 or 5 $\mu$m in diameter, and irradiated. UV irradiation was achieved using a UV lamp (SUV-16, As One, Japan) with a dose rate of 0.4 J/m$^2$·s, which was monitored with a UV radiometer (UVX Radiometer, UVP) at 254 nm, or with a UV box equipped with a UV-C lamp monitored with a VLX-3W radiometer (Vilber Lourmat) at 254 nm (CX-254 sensor). The filter was removed and cells were either fixed or cultured for the indicated time before fixation, and processed for immunofluorescence. For FRAP analysis, cells were cultured for 20 min to allow recruitment to sites of damage before photobleaching.

## IP from Cdt2-FLAG–expressing cells

Control or Cdt2-FLAG expressing 293 cells that were UV irradiated (50 J/m$^2$), or not UV irradiated, or in the middle S phase (5 h after release from aphidicolon arrest) were fixed with 0.02% formaldehyde for 10 min, lysed using 0.1% Triton X-100-containing modified cytoskeleton (mCSK) buffer (10 mM Pipes, pH 7.9, 100 mM NaCl, 300 mM sucrose, 0.1% [vol/vol] Triton X-100, 1 mM phenylmethylsulfonyl fluoride, 10 mM $\beta$-glycerophosphate, 1 mMNa$_3$VO$_4$, 10 mM NaF), and then sonicated. After centrifugation (120,000$g$, Beckman Type 45Ti rotor, for 20 min at 4°C), the supernatants were mixed with anti-FLAG antibody-conjugated magnetic beads (M8823; Sigma-Aldrich) for 1 h at 4°C to obtain immunoprecipitates. The precipitate was washed with ice-cold 0.1% Triton X-100 containing mCSK buffer and subsequently suspended in SDS sample buffer.

## RNAi knockdown experiments

Double-stranded RNAs were transfected at 100 nM using Oligofectamine (Invitrogen) or HiPerFect (QIAGEN). Twenty-four hours after the first transfection, a second transfection was performed and cells were cultured for two more days. The following siRNAs were made by Dharmacon: Skp2; GCAUGUACAGGUGGCUGUU, Cdt2_3′UTR; GCUGAGCUUUGGUCCACUA. The siRNA for siLuc, known as GL2, was used as a control siRNA. For Cdt1 silencing in Fig 1A and B, unsynchronized HeLa cells were transfected twice with 200 nM of Cdt1 siRNA or control luciferase siRNA using Lipofectamine 2000 with a time interval of 24 h and were analysed 48 h after the second transfection. In synchronized MCF7 cells (Fig 1D), a single transfection with 200 nM of Cdt1 siRNA or control luciferase siRNA using Lipofectamine 2000 5 h after release from thymidine and before adding nocodazole was carried out. The Cdt1 siRNA was made by MWG: AACGUGGAUGAAGUACCCGACTT.

## Chromatin fractionation

Cell extracts were prepared using 0.1% Triton X-100–containing mCSK buffer (200 $\mu$g of total protein in 400–800 $\mu$l). After centrifugation (17,900$g$ for 10 min at 4°C), soluble and chromatin-containing pellet fractions were obtained. The pellets were washed with ice-cold 0.1% Triton X-100–containing mCSK buffer and subsequently suspended in SDS sample buffer.

## Protein purification

### PCNA
PCNA was purified essentially as a trimer as described (Fukuda et al, 1995). Cell lysate was prepared from pT7-PCNA–transformed Rosetta (DE3) and sequentially subjected to Resource Q, Superdex 200, and Mono Q (GE Healthcare Life Science) column chromatography.

### Cdt1-3FLAG
The baculoviruses for Cdt1-3FLAG expression were infected into Sf21 insect cells, and cultured at 27°C for 60 h. Cell lysate was prepared with 0.5 M NaCl–containing buffer B (50 mM Tris–HCl, pH 8.0, 0.15 M NaCl, 1 mM EDTA, 10% glycerol, 1× Protease inhibitor cocktail [Roche Applied Science], 1 mM PMSF, 2 $\mu$g/ml Leupeptin), and passed on Diethylaminoethyl (DEAE) column. The flowthrough fraction was mixed with anti-FLAG resin (Sigma-Aldrich) and Cdt1-3FLAG was eluted with 200 $\mu$g/ml 3× FLAG peptides (Sigma-Aldrich).

### RFC complex
The baculoviruses for FLAG-Rfc1, Rfc2, Rfc3, Rfc4, and Rfc5 were co-infected to High Five cells, expressed, and purified as described (Shiomi et al, 2000).

### CRL4$^{Cdt2}$
The baculoviruses for Cdt2-3FLAG (WT or PIP-3A), DDB1, HA-Cul4, and His-myc-Rbx1 were co-infected into Sf21 cells, and purified as described (Hayashi et al, 2014). Briefly, the cell lysate was prepared with 0.5 M NaCl and 0.5% NonidetP-40–containing buffer B, clarified by centrifugation, and passed on a DEAE column. The flowthrough fraction was mixed with anti-FLAG resin and CRL4$^{Cdt2-3FLAG}$ was eluted with buffer B containing 0.1 M NaCl, 0.1% NP-40 and 200 $\mu$g/ml 3FLAG peptide (Sigma-Aldrich). If necessary, CRL4$^{Cdt2-3FLAG}$ was loaded on 15–35% glycerol gradient in buffer G (25 mM Hepes, pH 7.8, 0.1 M NaCl, 1 mM EDTA, 0.01% NP-40) and subjected to ultra-centrifugation (Beckman TLS-55) at 214,000 $g$ for 18 h at 4°C. 100 $\mu$l fractions were collected.

## In vitro binding assay for Cdt1 and CRL4$^{Cdt2}$ with PCNA

Cdt1-3FLAG (WT) and CRL4$^{Cdt2-3FLAG \ (WT)}$ beads were prepared as follows (Fig 5B): Cleared lysates were made from insect cells infected with baculoviruses for Cdt1-3FLAG (WT) or CRL4$^{Cdt2-3FLAG \ (WT)}$; incubated with anti-FLAG magnetic beads pre-blocked with buffer B containing 50% FBS, 0.5 M NaCl, and 0.5% NP-40; and washed with 0.5 M NaCl and 0.5% NP-40–containing buffer B and then with 0.1 M NaCl and 0.1% NP-40–containing buffer B. The protein beads prepared as above were incubated with PCNA protein at 4°C for 2 h, washed, and subjected to Western blotting.

## Preparation of ncDNA beads

Closed circular (cc) DNA beads were prepared as described (Higashi et al, 2012). Briefly, single-stranded DNA templates prepared using pBluescript II KS(–) were annealed with a biotinated oligonucleotide primer (5′-CGCCTTGATCGT [biotin-dT]GGGAACCGGAGCTGAAT-GAAGC-3′). After second-strand synthesis and ligation, covalently cc double-stranded DNA was purified by CsCl/EtBr density gradient centrifugation, incubated with streptavidin, and bound to biotin-sepharose beads to produce the cc DNA beads (around 100 ng per 1 $\mu$l bed volume of beads). A single nick was introduced by Nb.BbvCI (New England Biolabs) treatment to produce the ncDNA beads. As control beads, biotin-sepharose beads were incubated with streptavidin alone (sa-beads).

## PCNA loading and binding assay with purified Cdt1 and CRL[Cdt2]

For loading of PCNA on DNA beads, 2 $\mu$l bed volume of nc DNA beads were incubated with 280 ng of RFC and 200 ng of PCNA in 20 $\mu$l of 2mM ATP–containing HBS buffer (10 mM Hepes, pH 7.4, 10 mM MgCl$_2$, 0.2 mM EDTA, 0.05 % Tween 20, and 0.15 M NaCl) at 37°C for 1 h and washed with HBS buffer three times. The sa-beads were treated in the same way and used as control beads. The beads were incubated with Cdt1 or CRL4[Cdt2] alone at 4°C for 1 h and washed with HBS buffer. The bound proteins were analysed on Western blotting. To measure the amounts of nc DNA bound to beads, nc DNA was released from beads after incubation in a proteinase K buffer (10 mM Tris–HCl, pH 7.5, 50 mM NaCl, 1 mM EDTA, 0.75 % SDS, and 2 mg/ml proteinase K) at 50°C for 1 h, extracted with phenol chloroform and ethanol-precipitated. The purified nc DNA was run on 0.8% agarose gel in 1× TBE (Tris-borate-EDTA) buffer and detected with ethidium bromide.

## In vitro binding of PIP box peptides to PCNA

The expression plasmid for full-length human PCNA has been described (Hibbert & Sixma, 2012). Human PCNA was expressed in Bl21 (DE3) *Escherichia coli* cells (Merck KGaA) and purified as described (Hibbert & Sixma, 2012). The peptides Cdt2[PIP] (704–717), Cdt2[PIP]-TAMRA (704–717), Cdt1[PIP] (1–14), and Cdt1[PIP]-TAMRA (1–14) were synthesized in-house using Fmoc synthesis.

　FP was monitored using a PheraStar plate reader (BMG Labtech) in non-binding 96-well plates (Corning), using an excitation wavelength of 540 nm and an emission wavelength of 590 nm. Experiments were performed in a buffer containing 10 mM Hepes, pH 7.5, 125 mM NaCl, and 0.5 mM Tris (2-carboxyethyl) phosphine hydrochloride. In the direct binding experiments, 1 nM of labelled peptide was mixed with PCNA at final concentrations ranging from 0 to 2 $\mu$M. Error bars represent standard deviations from triplicate experiments. The data were fitted according to standard equations (Littler et al, 2010).

## X-ray crystallography structure determination

The purified PCNA was co-crystallized with Cdt2 and Cdt1 peptides, using the sitting drop vapour diffusion method in MRC 3-well crystallization plates (Swissci), with standard screening procedures (Newman et al, 2005). The protein solution at 17 mg·ml$^{-1}$ was pre-incubated with an equimolar amount of peptide, and 0.1 $\mu$l of this solution was mixed with 0.1 $\mu$l of reservoir solution and equilibrated against a 30 $\mu$l reservoir. Crystals for the Cdt2 complex were obtained in 10% (wt/vol) PEG 6000 and 100 mM MES/HCl, pH 6.0. Crystals for the Cdt1 complex were obtained in 15% (wt/vol) PEG 400, 100 mM calcium acetate, and 100 mM MES/HCl, pH 6.0. Crystals appeared at 4°C within 72 h. Crystals were briefly transferred to a cryo-protectant solution containing the reservoir solution and 25% (wt/vol) PEG200 and vitrified by dipping in liquid nitrogen.

## Data collection and structure refinement

X-ray data were collected on beamline ID29 at the European Synchrotron Radiation Facility. The images were integrated with XDS (Kabsch, 2010) and merged and scaled with AIMLESS (Evans, 2011). The starting phases were obtained by molecular replacement using PHASER (McCoy et al, 2007; Winn et al, 2011) with an available PCNA structure (PDB, 1AXC) as the search model. The models were built using COOT (Emsley et al, 2010) and refined with REFMAC (Murshudov et al, 2011) in iterative cycles. Model re-building and refinement parameter adjustment were performed in PDB-REDO (Joosten et al, 2014); homology-based hydrogen bond restraints (van Beusekom et al, 2018) were used at some stages of the procedure. The quality of the models was evaluated by MOLPROBITY (Chen et al, 2010). Data collection and refinement statistics are presented in Table 1. PISA, "Protein Interfaces, Surfaces and Assemblies" service PISA at the European Bioinformatics Institute (http://www.ebi.ac.uk/pdbe/prot_int/pistart.html) (Krissinel & Henrick, 2007) was used to calculate binding energies for Cdt2 PIP and Cdt1 PIP.

## FRAP experiments and data analysis

Cells (HeLa or MCF7) were plated on 35-mm glass-bottom dishes in phenol-red–free medium (Invitrogen). FRAP experiments were conducted as previously described (Xouri et al, 2007b; Giakoumakis et al, 2017). A Leica TCS SP5 microscope equipped with a 63x magnification 1.4 NA oil-immersion lens and FRAP booster were used. During FRAP, cells were maintained at 37°C and 5% CO$_2$. A defined circular region of interest with 2 $\mu$m diameter (ROI1) was placed in the nucleus and at the recruitment sites. GFP was excited using a 488-nm argon laser line. 50 pre-bleach images were acquired with 3% of the 488-nm line at 60% argon laser intensity, followed by a single bleach pulse on ROI1 using 476-nm and 488-nm laser lines combined at maximum power. In this manner, at least 60% of the fluorescence in ROI1 was bleached successfully. Next, 400 post-bleach images with a 0.052 s interval were recorded. Mean intensities of ROI1, the whole nucleus (ROI2) and an area outside of the nucleus (ROI3) were quantified and exported as .csv format files. Data analysis was performed using easyFRAP (Rapsomaniki et al, 2012; Giakoumakis et al, 2017; Koulouras et al, 2018).

# Supplementary Information

# Life Science Alliance

## Acknowledgements

This work was financially supported by JSPS KAKENHI grant numbers JP25131718 and JP26291025 (to H Nishitani), 13J07320 (to A Hayashi), the European Research Council (ERC-StG 281851 and ERC-PoC 755284), the project Bioimaging-GR, MIS 5002755 under the Operational Program "Competitiveness, Enterpreneurship, and Innovation" co-financed by Greece and the EU (to Z Lygerou) and a Greek state scholarship (to A Perrakis). We thank the Advanced Light Microscopy Facility of the University of Patras.

### Author Contributions

A Hayashi: data curation, formal analysis, funding acquisition, and investigation.
NN Giakoumakis: data curation, formal analysis, and investigation.
T Heidebrecht: data curation, formal analysis, and investigation.
T Ishii: data curation, formal analysis, and investigation.
A Panagopoulos: data curation, formal analysis, and investigation.
C Caillat: data curation, formal analysis, and investigation.
M Takahara: data curation, formal analysis, and investigation.
RG Hibbert: data curation, formal analysis, and investigation.
N Suenaga: data curation, formal analysis, and investigation.
M Stadnik-Spiewak: data curation, formal analysis, and investigation.
T Takahashi: methodology.
Y Shiomi: validation and investigation.
S Taraviras: validation and investigation.
E von Castelmur: validation and investigation.
Z Lygerou: supervision, funding acquisition, investigation, and writing—review and editing.
A Perrakis: supervision, funding acquisition, investigation, and writing—original draft.
H Nishitani: supervision, funding acquisition, investigation, and writing—review and editing.

### Conflict of Interest Statement

The authors declare that they have no conflict of interest.

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
