## [Reviewer comments · Life Science Alliance]

Life Science Alliance

Direct binding of Cdt2 to PCNA is important for targeting the CRL4Cdt2 E3 ligase activity to Cdt1

Akiyo Hayashi, Nickolaos Giakoumakis, Tatjana Heidebrecht, Takashi Ishii, Andreas Panagopoulos, Christophe Caillat, Michiyo Takahara, Richard Hibbert, Naohiro Suenaga, Magda Stadnik-Spiewak, Tatsuro Takahashi, yasushi Shiomi, Stavros Taraviras, Eleonore von Castelmur, Zoi Lygerou, Anastassis Perrakis, and Hideo Nishitani

DOI: [10.26508/lsa.201800238](https://doi.org/10.26508/lsa.201800238)

Corresponding author(s): *Hideo Nishitani, University of Hyogo*

Review Timeline:

Submission Date:	2018-11-11
Editorial Decision:	2018-12-05
Revision Received:	2018-12-13
Editorial Decision:	2018-12-14
Revision Received:	2018-12-17
Accepted:	2018-12-17

Scientific Editor: Andrea Leibfried

Transaction Report:

December 5, 2018

RE: Life Science Alliance Manuscript #LSA-2018-00238-T

Prof. Hideo Nishitani
University of Hyogo
Graduate School of Life Science
Kouto 3-2-1
Kamigori
Ako-gun, Hyogo 678-1297
Japan

Dear Dr. Nishitani,

Thank you for submitting your revised manuscript entitled "Direct binding of Cdt2 to PCNA is important for targeting the CRL4Cdt2 E3 ligase activity to Cdt1". It has been reviewed by three experts, and I enclose their reports below.

As you will see, the reviewers appreciate your data and provide constructive input on how to further strengthen your manuscript. The changes required are rather minor, and I would thus like to invite you to provide a point-by-point response and a revised version of your manuscript. All concerns raised can be addressed by changes to the manuscript text and discussion or by changing the presentation of the data. No additional experiments are needed. Please note that it is important to include the PDB identifiers and to improve the crystallographic part as outlined by reviewer #3.

A. FINAL FILES:

-- High-resolution figure, supplementary figure and video files uploaded as individual files: See our detailed guidelines for preparing your production-ready images, <http://life-science-alliance.org/authorguide>

-- Summary blurb (enter in submission system): A short text summarizing in a single sentence the study (max. 200 characters including spaces). This text is used in conjunction with the titles of papers, hence should be informative and complementary to the title. It should describe the context and significance of the findings for a general readership; it should be written in the present tense

and refer to the work in the third person. Author names should not be mentioned.

B. MANUSCRIPT ORGANIZATION AND FORMATTING:

Full guidelines are available on our Instructions for Authors page, <http://life-science-alliance.org/authorguide>

Sincerely,

Reviewer #1 (Comments to the Authors (Required)):

The manuscript by Hayashi et al. is a very high quality study that definitively shows that Cdt2, the

substrate receptor for the CRL4-Cdt2 ubiquitin ligase complex, binds to PCNA independently of substrates to target PCNA-bound substrates. CRL4-Cdt2 is a key regulator of genome stability, in part through the degradation of the replication licensing factor Cdt1. It was known that CRL4-Cdt2 recognized Cdt1 based on a PIP-box sequence in Cdt1 that made Cdt1 localize to chromatin-loaded PCNA - either upon DNA damage or during S phase - with only the PCNA-bound form of Cdt1 targeted for ubiquitylation. This study shows that CRL4-Cdt2 is independently loaded onto one of the trimeric subunits of PCNA prior to Cdt1 binding to PCNA. The authors identify a PIP-box sequence in the C-terminus of Cdt2, and mutate it to show that it is required for the interaction with Cdt1 and for Cdt1 degradation. As other substrates of CRL4-Cdt2 appear to be targeted via the same mechanism (including the CDK inhibitor p21, which the authors also study), this mechanism will have broader implications for the targeting of substrates in response to DNA replication. Additionally, the use of a multimeric protein as a scaffold to bring a ubiquitin ligase and its substrates together is a new paradigm for the regulation of ubiquitin-mediated substrate degradation.

The experimental data is very well controlled and compelling. In particular, the inclusion of crystal structure data for the Cdt2-PIP box sequence bound to PCNA is very useful. There is another paper that has been accepted that overlaps with some of this information, which the authors acknowledge. However, this study by Hayashi et al. is much more thorough and comprehensive, including the crystal structure analysis, and so the presence of a co-published study that partially overlaps should not affect the acceptance of this manuscript, which I strongly support. There are only a few minor points that I trust the authors to address.

Minor points:

1. Figure 2 legend: panel C needs to be listed with a "C" in the legend.
2. Figure 3B is not compelling. First, it is impossible to see the IP level of FLAG-Cdt2 (1-700) because of close antibody heavy chain band signal. Second, there is less FLAG-Cdt2 (1-700) in the whole-cell lysate - and there is less PCNA signal. It does not appear that there is a substantial difference in the PCNA signal when divided by starting material. Quantitation of the ratio - or repeating the experiment with more equivalent expression would be useful. The subsequent experiments using PIP-3A suggests strongly that this result is correct.
3. Figure 3D legend: The legend says there is an asterisk label for 3D, but it is not shown.
4. Figure 4A: It would be easier to follow the data if Cdt2-PIP-3A also had a line curve fitted - as it is hard to see where the lightly-colored symbols are on the graph without a line. Or alternatively, the symbols could be filled in to make them easier to see.
5. Figure 4: If it is available, it would be helpful to add another panel showing the full-length PCNA crystal structure with Cdt1-PIP and Cdt2-PIP peptides bound, so that the reader can see where the binding is on the full PCNA. Currently, only zoomed-in images are shown in panels A and B and C that show only the small part of PCNA that is bound to the peptides.
6. Figure 7D legend: It would be helpful to provide y-axis labels.

Reviewer #2 (Comments to the Authors (Required)):

This paper clarifies the mode of action of CRL4-Cdt2 ubiquitin ligase by showing that the Cdt2

substrate receptor can directly interact with PCNA, and co-interaction of CRL4-Cdt2 and Cdt1 with different subunits of the PCNA trimer may facilitate ubiquitylation by bringing ligase and substrate into proximity. The data are clear and generally clearly presented and I have only minor comments. Note that similar findings have recently been published by Leng et al (this paper is cited).

1. It would be interesting to discuss the possible biological significance of the difference in affinity shown by Cdt2 and Cdt1 for PCNA.

2. The data suggest that Cdt2 interacts more strongly with PCNA-DNA than with PCNA that is not associated with DNA, even though the Cdt2 PIP peptide has a high affinity for PCNA. Reference is made to Ivanov et al., suggesting a conformational change in PCNA on DNA binding may be involved, but it would be of interest to discuss this at greater length, since it is key to understanding how CRL4-Cdt2 substrate degradation is coordinated with DNA replication or repair.

Minor comments:

1. Some graphs lack error bars e.g. Fig. 1B & 2C; it is sufficient to show the range of two experiments.

Reviewer #3 (Comments to the Authors (Required)):

This manuscript describes molecular mechanisms of Cdt2-PCNA and Cdt1-PCNA interactions, and CRL4(Cdt2) activity. The authors identified PIP (PCNA-interacting protein motif) in the C-terminal region of Cdt2. That finding was supported by crystallography and biochemistry. Furthermore, the authors provided a plausible model, where co-localization onto PCNA brings the E3 ligase and its substrate in proximity. The proposed mechanism is so exciting. However, added experiments and explanation could better support the authors conclusions.

Base on FP experiments, the authors describe that Cdt2PIP is tightly bound to PCNA comparable to p21. However, p21 peptide used in this paper (residues 142-156 of p21) is shorter than the commonly used p21 peptides (residues 139-160 or 141-160). The p21 peptide in this paper lacks the C-terminal four residues, LIFS. The four residues are crucially involved in the interaction between p21 and PCNA (Gulbis et al., Cell, 1996) and the truncation of LIFS causes about 15 times reduction of inhibition activity compared to a p21 peptide (residues 141-160) (Zheleva et al., Biochemistry, 2000). So, FP experiment using p21 peptide (residues 139-160 or 141-160) is required to emphasize tight binding of Cdt2PIP to PCNA.

FP experiments show that about 100 times tighter binding of Cdt2PIP to PCNA compared to Cdt1PIP. However, amounts of input and beads in Fig5C imply that the interaction of Cdt2PIP with PCNA is rather weaker than the interaction of Cdt1PIP with PCNA. Other report gives similar impression (Leng et al., JBC, 2018).

Conserved sequence of canonical PIP is Qxx(ψ)xx(φ)(φ), where ψ is hydrophobic residue with blanch side-chain and φ is aromatic residue. Cdt1PIP is canonical PIP, because it has Q. In contrast, Cdt2PIP has M instead of Q. Such kind of PIP seems to have lower affinity for PCNA compared to canonical PIP (Hishiki et al., 2009). It will be convincing if the author give some structural-based explanation or discussion about high affinity of Cdt2 for PCNA. Are there additional interactions of Cdt2PIP with PCNA to stabilize the binding?

Alternatively, is it likely that FP data for Cdt2PIP and Cdt1PIP were other way around?

Display of crystallographic part is too crude to publish. There are quite a few points to be improved. In table 1, "PDB identifiers" are not shown. The authors should deposit structural data to Protein Data Bank, get PDB entry codes (identifiers), and describe them in the paper. "PCNA/Cdt2 peptide in A.U." should be "PCNA monomer/peptide in A.U.". "Atoms protein" should be "Protein atoms". "B-factors protein" should be "Averaged B-factors". Values of rmsZ and rmsd in bond lengths and angles would be other way around. Figure legends of 4B and 4C are also other way around. The authors should show labels for amino acid residues in Figs. 4B and 4C. The authors show electron density maps for Cdt1 and Cdt2 peptides in Fig.S2. The viewing directions are different from those of Figs. 4B and 4C. For easiness, similar directions are preferable.

The authors described binding energies for Cdt2PIP and Cdt1PIP. The authors should also describe the way to estimate those values.

Reviewer #1 (Comments to the Authors (Required)):

We thank this reviewer for her/his positive evaluation on our manuscript and helpful suggestions and comments.

The manuscript by Hayashi et al. is a very high quality study that definitively shows that Cdt2, the substrate receptor for the CRL4-Cdt2 ubiquitin ligase complex, binds to PCNA independently of substrates to target PCNA-bound substrates. CRL4-Cdt2 is a key regulator of genome stability, in part through the degradation of the replication licensing factor Cdt1. It was known that CRL4-Cdt2 recognized Cdt1 based on a PIP-box sequence in Cdt1 that made Cdt1 localize to chromatin-loaded PCNA - either upon DNA damage or during S phase - with only the PCNA-bound form of Cdt1 targeted for ubiquitylation. This study shows that CRL4-Cdt2 is independently loaded onto one of the trimeric subunits of PCNA prior to Cdt1 binding to PCNA. The authors identify a PIP-box sequence in the C-terminus of Cdt2, and mutate it to show that it is required for the interaction with Cdt1 and for Cdt1 degradation. As other substrates of CRL4-Cdt2 appear to be targeted via the same mechanism (including the CDK inhibitor p21, which the authors also study), this mechanism will have broader implications for the targeting of substrates in response to DNA replication. Additionally, the use of a multimeric protein as a scaffold to bring a ubiquitin ligase and its substrates together is a new paradigm for the regulation of ubiquitin-mediated substrate degradation.

The experimental data is very well controlled and compelling. In particular, the inclusion of crystal structure data for the Cdt2-PIP box sequence bound to PCNA is very useful. There is another paper that has been accepted that overlaps with some of this information, which the authors acknowledge. However, this study by Hayashi et al. is much more thorough and

comprehensive, including the crystal structure analysis, and so the presence of a co-published study that partially overlaps should not affect the acceptance of this manuscript, which I strongly support. There are only a few minor points that I trust the authors to address.

Minor points:

1. Figure 2 legend: panel C needs to be listed with a "C" in the legend.

We listed with "C" in the figure legend.

2. Figure 3B is not compelling. First, it is impossible to see the IP level of FLAG-Cdt2 (1-700) because of close antibody heavy chain band signal. Second, there is less FLAG-Cdt2 (1-700) in the whole-cell lysate - and there is less PCNA signal. It does not appear that there is a substantial difference in the PCNA signal when divided by starting material. Quantitation of the ratio - or repeating the experiment with more equivalent expression would be useful. The subsequent experiments using PIP-3A suggests strongly that this result is correct.

We apologize for the poor detection of FLAG-Cdt2(1-700). We replaced the old western with another one, which showed clearly the FLAG-Cdt2(1-700) band signal. FLAG-Cdt2(1-730) and FLAG-Cdt2(1-700) in the lysate and immunoprecipitation are detected at almost similar levels.

3. Figure 3D legend: The legend says there is an asterisk label for 3D, but it is not shown.

We showed asterisk (*) in the Figure 3D, which indicate bands derived from immunoglobulin.

4. Figure 4A: It would be easier to follow the data if Cdt2-PIP-3A

also had a line curve fitted - as it is hard to see where the lightly-colored symbols are on the graph without a line. Or alternatively, the symbols could be filled in to make them easier to see.

We made the symbols for that line thicker and darker, so they are more easily visible in the revised one.

5. Figure 4: If it is available, it would be helpful to add another panel showing the full-length PCNA crystal structure with Cdt1-PIP and Cdt2-PIP peptides bound, so that the reader can see where the binding is on the full PCNA. Currently, only zoomed-in images are shown in panels B and C that show only the small part of PCNA that is bound to the peptides.

Figure 4 has been modified accordingly, and full PCNA crystal structure with Cdt1 PIP peptides and Cdt2 PIP peptides are shown in new Figure 4B and 4C. We are sorry but the crystal structure with Cdt1-PIP and Cdt2-PIP were oppositely shown in old Figure 4B and 4C. We corrected them in the new Figure 4B and 4C.

6. Figure 7D legend: It would be helpful to provide y-axis labels.

We added y-axis labels (Cdt1+ cells (%)) in Figure 7D.

Reviewer #2 (Comments to the Authors (Required)):

We thank this reviewer for thoughtful and valuable comments, which helped us to improve our manuscript.

This paper clarifies the mode of action of CRL4-Cdt2 ubiquitin ligase by showing that the Cdt2 substrate receptor can directly interact with PCNA, and co-interaction of CRL4-Cdt2 and Cdt1 with different subunits of the PCNA trimer may facilitate ubiquitylation by bringing ligase and substrate into proximity.

The data are clear and generally clearly presented and I have only minor comments. Note that similar findings have recently been published by Leng et al (this paper is cited).

1. It would be interesting to discuss the possible biological significance of the difference in affinity shown by Cdt2 and Cdt1 for PCNA.

Concerning to what the biological significance of difference in affinity of Cdt2 and Cdt1 to PCNA is and how Cdt2-PIP contributes to promote Cdt1 degradation, we discussed them in more detail in Discussion on page 21 as follows. We also included a model illustrating to explain them, shown as a new Supplementary Figure S9.

".....Here, we propose a synergistic mechanism where the Cdt2 PIP-box and the Cdt1 PIP-degron operate in concert to promote CRL4^{Cdt2}-dependent target ubiquitination (Figure 9, **Supplementary Figure S9**). It is also noteworthy that the binding of Cdt1 onto PCNA is transient in cells, while Cdt2 interactions are more stable (Roukos et al., 2011 and Figure 1B): a synergistic mechanism would imply that interactions of pre-bound Cdt2 with transiently formed Cdt1 PIP-degron, are important for localising CRL4^{Cdt2} and Cdt1 onto PCNA. **The property that affinity of Cdt2 PIP-box to PCNA is higher than that of Cdt1 PIP-box (Figure 4A) is compatible with such a model. In addition, weaker affinity of Cdt1 PIP-box can help to release Cdt1 after poly-ubiquitination, while CRL4^{Cdt2} could remain on PCNA to trap next substrate, leading to an efficient degradation cycle of substrates (Supplementary Figure S9). In the absence of Cdt2 PIP-box, the process of ubiquitination needs to be performed sequentially; transient Cdt1 binding to PCNA and recognition by Cdt2 via its**

N-terminal WD40 repeats, which would be, however, less effective for Cdt1 degradation.”

2. The data suggest that Cdt2 interacts more strongly with PCNA-DNA than with PCNA that is not associated with DNA, even though the Cdt2 PIP peptide has a high affinity for PCNA. Reference is made to Ivanov et al., suggesting a conformational change in PCNA on DNA binding may be involved, but it would be of interest to discuss this at greater length, since it is key to understanding how CRL4-Cdt2 substrate degradation is coordinated with DNA replication or repair.

This is a very important comment referring to the basis of control on CRL4-Cdt2 ubiquitin ligase that functions only when PCNA is loaded on DNA. The change in the PCNA structure on DNA is one explanation. In addition, we discussed possible mechanism in Discussion on page 21 as follows,

“Our biochemical analysis demonstrated that CRL4^{Cdt2} interacts more strongly with PCNA^{on DNA} than with PCNA that is not associated with DNA (Fig.5B and 5C). Since Cdt2 PIP peptide can bind to free PCNA (Fig. 4), there is a mechanism that enhances the affinity of CRL4^{Cdt2} to the PCNA^{on DNA}. This is compatible with models that suggest a conformational change upon DNA binding can modulate affinity. It is tempting to speculate that the high affinity CRL4^{Cdt2} complex docking into DNA bound PCNA, triggers changes that allow the transient loading of low affinity substrates like Cdt1, enabling rapid ubiquitination molecules in the vicinity. On the other hand, we reproducibly observed that Cdt2 was recovered, though at lower levels, on DNA-beads in the absence of PCNA (Figure 5C), suggesting that CRL4^{Cdt2} has a DNA binding activity. Furthermore, RFC1 complex may have a role connecting PCNA and Cdt2, as Cdt2 PIP-3A was detected to some more levels in the presence of PCNA and

RFC1-complex (Figure 5D). As reported, PCNA loaders appear to have an additional role in the CRL4^{Cdt2} mediated ubiquitination after loading PCNA (Shiomi et al., 2012). The extended C-terminal region of Cdt2 might have an additional domain involved in such a regulation, or unknown factor might be involved. The exact molecular mechanisms that interplay to regulate these interactions need to be investigated further.”

Minor comments:

1. Some graphs lack error bars e.g. Fig. 1B & 2C; it is sufficient to show the range of two experiments.

We added error bars in Fig. 1B and 2C in new Figures.

Reviewer #3 (Comments to the Authors (Required)):

We thank this reviewer for his/her careful reading and valuable comments. They helped to improve our manuscript.

This manuscript describes molecular mechanisms of Cdt2-PCNA and Cdt1-PCNA interactions, and CRL4(Cdt2) activity. The authors identified PIP (PCNA-interacting protein motif) in the C-terminal region of Cdt2. That finding was supported by crystallography and biochemistry. Furthermore, the authors provided a plausible model, where co-localization onto PCNA brings the E3 ligase and its substrate in proximity. The proposed mechanism is so exciting. However, added experiments and explanation could better support the authors conclusions.

Base on FP experiments, the authors describe that Cdt2PIP is tightly bound to PCNA comparable to p21. However, p21 peptide used in this paper (residues 142-156 of p21) is shorter than the commonly used p21 peptides (residues 139-160 or 141-160).

The p21 peptide in this paper lacks the C-terminal four residues, LIFS. The four residues are crucially involved in the interaction between p21 and PCNA (Gulbis et al., Cell, 1996) and the truncation of LIFS causes about 15 times reduction of inhibition activity compared to a p21 peptide (residues 141-160) (Zheleva et al., Biochemistry, 2000). So, FP experiment using p21 peptide (residues 139-160 or 141-160) is required to emphasize tight binding of Cdt2PIP to PCNA.

The referee is right that longer peptides of P21 might bind tighter than the peptide we use. However, we compared the binding affinity between same length of PIP peptides with PIP-box at a same position, and our main point remains the comparison of Cdt1 and Cdt2 PIPs. The biological significance of difference in affinity of Cdt2-PIP and Cdt1-PIP to PCNA is more thoroughly discussed in the revised manuscript on page 21.

FP experiments show that about 100 times tighter binding of Cdt2PIP to PCNA compared to Cdt1PIP. However, amounts of input and beads in Fig5C imply that the interaction of Cdt2PIP with PCNA is rather weaker than the interaction of Cdt1PIP with PCNA. Other report gives similar impression (Leng et al., JBC, 2018).

Bead based experiments could suffer from many artefacts and do not provide quantitative measurements but qualitative. We believe that our quantitation, is correct, under the experimental conditions we use.

In our Binding assay condition of Figure 5C, 148 fmol or 158 fmol of PCNA trimer were loaded on one plasmid DNA on bead for Cdt1 or Cdt2 binding assay, respectively. That corresponded roughly three PCNA trimers were loaded on one plasmid DNA. In this condition, 310 fmol Cdt1 and 408 fmol Cdt2 were detected as bound on PCNA. This means that 2.1 Cdt1 molecules are bound to one PCNA trimer ($301 \text{ fmol Cdt1} / 148 \text{ fmol PCNA trimer} = 2.0$), and 2.6 Cdt2 molecules are bound to one PCNA trimer ($408 \text{ fmol Cdt2} / 158 \text{ fmol PCNA trimer} = 2.6$), slightly higher amounts than

Cdt1. This indicate that more than two parts of each PCNA trimer were occupied by Cdt1 or Cdt2, suggesting that most of PIP-box acceptor sites on PCNA were bound and saturated with Cdt1 or CRL4Cdt2 in our binding assay condition. We think that when assayed with increased salt conditions or lower protein levels, a binding difference between Cdt1 and Cdt2 to beads could be detected. In addition, the phosphorylation on Cdt2 could affect binding assay. As we reported, Cdt2 phosphorylation likely affects its activity and binding to PCNA (Sakaguchi et al, 2012; Rizzardi et al, 2015; Nukina et al, 2018). We noticed that our preparation of Cdt2 from insect cells were phosphorylated. Therefore, it is probable that that phosphorylation on Cdt2 affected the binding assay. We included a data showing that our Cdt2 preparation of CRL4 complex was phosphorylated, verified by phosphatase treatment, as a new Supplementary Figure S4D, and mentioned in the figure legend. We discussed a possible binding assay using phosphorylated and un-phosphorylated Cdt2 to address how phosphorylation contributes to PCNA binding in Discussion on page 23.

“We noticed that our Cdt2 preparation from insect cells was phosphorylated (Supplementary Figure 4D) at levels found in human cells. The assay with de-phosphorylated Cdt2 and kinase-treated Cdt2 could help to understand how the phosphorylation on Cdt2 regulate its binding activity.”

Conserved sequence of canonical PIP is Qxx(psi)xx(phi) (phi), where psi is hydrophobic residue with blanch side-chain and phi is aromatic residue. Cdt1PIP is canonical PIP, because it has Q. In contrast, Cdt2PIP has M instead of Q. Such kind of PIP seems to has lower affinity for PCNA compared to canonical PIP (Hishiki et al., 2009). It will be convincing If the author give some structural-based explanation or discussion about high affinity of Cdt2 for PCNA. Are there additional interactions of Cdt2PIP with PCNA to stabilize the binding?

Alternatively, is it likely that FP data for Cdt2PIP and Cdt1PIP were other way around?

Yes, as discussed in the paper in page 12 "the average buried area upon the binding of the Cdt2 peptide is $692 \pm 6 \text{ \AA}^2$ and average calculated energy of binding is $-13 \pm 0.2 \text{ kcal mol}^{-1}$ while the average buried area upon the binding of the Cdt1 peptide is $650 \pm 4 \text{ \AA}^2$ and average calculated energy of binding is $-8.3 \pm 0.5 \text{ kcal mol}^{-1}$; these confirm the tighter binding of the Cdt2 peptide to PCNA". The structural differences explain the difference in affinity.

Display of crystallographic part is too crude to publish. There are quite a few points to be improved. In table 1, "PDB identifiers" are not shown. The authors should deposit structural data to Protein Data Bank, get PDB entry codes (indentifiers), and describe them in the paper.

The PDB codes will be supplied. So, we are sorry, but in the current manuscript, we tentatively wrote the crystal structural PDB codes for PCNA bound with Cdt1-PIP peptide and Cdt2-PIP peptide as XXX and YYY, respectively.

"PCNA/Cdt2 peptide in A.U." should be "PCNA monomer/peptide in A.U.". "Atoms protein" should be "Protein atoms". "B-factors protein" should be "Averaged B-factors". Values of rmsZ and rmsd in bond lengths and angles would be other way around.

All these were corrected. We thank the reviewer for looking thoroughly and apologize for this.

Figure legends of 4B and 4C are also other way around. The authors should show labels for amino acid residues in Figs. 4B and 4C.

We apologize for the mistake for figure legends. 4B and 4C were oppositely listed. We corrected them and labeled for amino acid residues in the new figures.

The authors show electron density maps for Cdt1 and Cdt2 peptides in Fig.S2. The viewing directions are different from those of Figs. 4B and 4C. For easiness, similar directions are preferable.

The direction in which the binding mode is clear, does not allow to show the density well, and vice versa.

We appreciate the comment, but the two views were selected to demonstrate the binding and the density in the best possible manner.

The authors described binding energies for Cdt2PIP and Cdt1PIP. The authors should also describe the way to estimate those values.

We apologize for omitting the reference to PISA.

'Protein interfaces, surfaces and assemblies' service PISA at the European Bioinformatics Institute.

(http://www.ebi.ac.uk/pdbe/prot_int/pistart.html),
[paper]E. Krissinel and K. Henrick (2007). 'Inference of macromolecular assemblies from crystalline state.'. J. Mol. Biol. 372, 774--797. We mentioned it in the Experimental procedure and the reference was cited.

December 14, 2018

RE: Life Science Alliance Manuscript #LSA-2018-00238-TR

Prof. Hideo Nishitani
University of Hyogo
Graduate School of Life Science
Kouto 3-2-1
Kamigori
Ako-gun, Hyogo 678-1297
Japan

Dear Dr. Nishitani,

Thank you for submitting your revised manuscript entitled "Direct binding of Cdt2 to PCNA is important for targeting the CRL4Cdt2 E3 ligase activity to Cdt1". I appreciate the introduced changes and would be happy to publish your paper in Life Science Alliance pending final revisions necessary to meet our formatting guidelines.

Please add a callout in the manuscript text to figure panel 2C and please add the PDB identifiers to the text. The latter needs to be done at the latest at proof stage, but please note that for a 2018 publication, returning the proofs needs to be really rapid. So inclusion of the PDB identifiers in the revised version would be better. We would need the final version of your manuscript by Monday morning to allow for publication this year.

A. FINAL FILES:

-- High-resolution figure, supplementary figure and video files uploaded as individual files: See our detailed guidelines for preparing your production-ready images, <http://life-science-alliance.org/authorguide>

-- Summary blurb (enter in submission system): A short text summarizing in a single sentence the study (max. 200 characters including spaces). This text is used in conjunction with the titles of papers, hence should be informative and complementary to the title. It should describe the context and significance of the findings for a general readership; it should be written in the present tense

and refer to the work in the third person. Author names should not be mentioned.

B. MANUSCRIPT ORGANIZATION AND FORMATTING:

Full guidelines are available on our Instructions for Authors page, <http://life-science-alliance.org/authorguide>

Thank you for your attention to these final processing requirements.

Sincerely,

December 17, 2018

RE: Life Science Alliance Manuscript #LSA-2018-00238-TRR

Prof. Hideo Nishitani
University of Hyogo
Graduate School of Life Science
Kouto 3-2-1
Kamigori
Ako-gun, Hyogo 678-1297
Japan

Dear Dr. Nishitani,

Thank you for submitting your Research Article entitled "Direct binding of Cdt2 to PCNA is important for targeting the CRL4Cdt2 E3 ligase activity to Cdt1". It is a pleasure to let you know that your manuscript is now accepted for publication in Life Science Alliance. Congratulations on this interesting work.

*****IMPORTANT:** If you will be unreachable at any time, please provide us with the email address of an alternate author. Failure to respond to routine queries may lead to unavoidable delays in publication.*******

DISTRIBUTION OF MATERIALS:

Again, congratulations on a very nice paper. I hope you found the review process to be constructive and are pleased with how the manuscript was handled editorially. We look forward to future exciting

submissions from your lab.

Sincerely,
